



# Dust transport and advection measurement with spaceborne lidars ALADIN, CALIOP and model reanalysis data

Guangyao Dai[1], Kangwen Sun[1], Xiaoye Wang[1], Songhua Wu[1,2,3], Xiangying E[1], Qi Liu[1], Bingyi Liu[1,2]

[1] Department of Marine Technology, College of Information Science and Engineering, Ocean University of China, Qingdao, 266100, China
[2] Laboratory for Regional Oceanography and Numerical Modelling, Pilot National Laboratory for Marine Science and Technology (Qingdao), Qingdao, 266200, China
[3] Institute for Advanced Ocean Study, Ocean University of China, Qingdao, 266100, China

*Correspondence to*: Songhua Wu (wush@ouc.edu.cn)

**Abstract.** In this paper, a long-term large-scale Sahara dust transport event which occurred between 14 June and 27 June 2020 is tracked with the spaceborne lidars ALADIN and CALIOP together with ECMWF and HYSPLIT analysis. We evaluate the performance of ALADIN and CALIOP on the observations of dust optical properties and wind fields and explore the possibility of tracking the dust events and calculating the dust mass advection with the combination of satellite and model data. The dust plumes are identified with AIRS/Aqua "Dust Score Index" and with the "Vertical Feature Mask" products from CALIOP. The emission, dispersion, transport and deposition of the dust event are monitored using the data from AIRS/Aqua, CALIOP and HYSPLIT. With the quasi-synchronized observations by ALADIN and CALIOP, combined with the wind field and relative humidity, the dust advection values are calculated. From this study, it is found that the dust event generated on 14 and 15 June 2020 from Sahara Desert in North Africa dispersed and moved westward over the Atlantic Ocean, finally being deposited in the Atlantic Ocean, the Americas and the Caribbean Sea. During the transport and deposition processes, the dust plumes are trapped in the Northeasterly Trade-wind zone between the latitudes of $5\,°N$ and $30\,°N$, and altitudes of 0 km and 6 km (in this paper we name this space area as the "Saharan dust westward transport tunnel"). From the measurement results on 19 June 2020, influenced by the hygroscopic effect and mixing with other types of aerosols, the backscatter coefficients of dust plumes were increasing along the transport routes, with $3.88\times10^{-6}\pm2.59\times10^{-6}\ m^{-1}sr^{-1}$ in the "dust portion during emission phase", $7.09\times10^{-6}\pm3.34\times10^{-6}\ m^{-1}sr^{-1}$ in the "dust portion during development phase" and $7.76\times10^{-6}\pm3.74\times10^{-6}\ m^{-1}sr^{-1}$ in the "dust portion during deposition phase". Finally, the advection value for different dust parts and heights on 19 June and on the entire transport routine during transportation are computed. On 19 June, the mean dust advection values are about $2.06\ mg\cdot m^{-2}\cdot s^{-1}$ in the dust portion during the emission phase, $1.47\ mg\cdot m^{-2}\cdot s^{-1}$ in the dust portion during the development phase and $0.95\ mg\cdot m^{-2}\cdot s^{-1}$ in the dust portion during the deposition phase. In the whole life-time of the dust event, the mean dust advection values were about $1.50\ mg\cdot m^{-2}\cdot s^{-1}$ on 15 June 2020, $2.41\ mg\cdot m^{-2}\cdot s^{-1}$ on 16 June 2020, $1.47\ mg\cdot m^{-2}\cdot s^{-1}$ on 19 June 2020,



$2.01 \, \mathrm{mg \cdot m^{-2} \cdot s^{-1}}$ on 24 June 2020 and $1.15 \, \mathrm{mg \cdot m^{-2} \cdot s^{-1}}$ on 27 June 2020. During the dust development stage, the mean advection values gradually increased and reaching their maximum on 16 June with the enhancement of the dust event. Then, the mean advection values decreased during the transport and the deposition of the dust over the Atlantic Ocean, the Americas and the Caribbean Sea.

## 1 Introduction

The global aerosol distribution and wind profiles have significant impacts on the atmospheric circulation, marine–atmosphere circulation and aerosol activities. As the most abundant aerosol types in the global atmosphere, the mineral dust influences the radiation budget, air quality, climate and weather via direct and various indirect radiative effects. Mineral dust is also considered as a major source of nutrients for ocean and terrestrial ecosystems. By the prevailing wind systems, mineral dust deposited over the ocean and land surface can significantly affect the carbon cycle and perturb the ocean and land geochemistry (Velasco-Merino et al., 2018; Banerjee et al., 2019). The atmospheric mineral dust can be transported over tens of thousands of kilometers away from its source regions (Uno et al., 2009, Haarig et al., 2017, Hofer et al., 2017). For instance, the biggest dust source, Africa, produced over half the global total dust (Huneeus et al., 2011), and African dust is transported westward over the Atlantic Ocean to reach South America (Yu et al., 2015; Prospero et al., 2020), the Caribbean Sea (Prospero and Lamb, 2003) and southern United States (Bozlaker et al., 2013). Hence, continuous observations of the dust long-range transport are crucial. As one of the best techniques for remotely studying the characteristics and properties of aerosols, lidar contributes much to measuring the dust distribution. As introduced in previous papers, several comprehensive field campaigns including Aerosol Characterization Experiment ACE-Asia (Huebert et al., 2003; Shimizu et al., 2004), the Puerto Rico Dust Experiment PRIDE (Colarco et al., 2003; Reid et al., 2003), the Saharan Dust Experiment SHADE (Tanré et al., 2003), the Saharan Mineral Dust Experiments SAMUM-1 (Heintzenberg, 2009) and SAMUM-2 (Ansmann et al., 2011), the Dust and Biomass-burning Experiment DABEX (Haywood et al., 2008), the Dust Outflow and Deposition to the Ocean project DODO (McConnell et al., 2008), the Pacific Dust Experiment PACDEX (Huang et al., 2008), the China-US joint dust field experiment (Huang et al., 2010), the Saharan Aerosol Long-Range Transport and Aerosol-Cloud-Interaction Experiment SALTRACE (Weinzierl et al., 2017), the study of Saharan Dust Over West Africa SHADOW, and the Central Asian Dust Experiment CADEX (Hofer et al., 2017, 2020a, 2020b) were conducted.

However, the measurement data from these campaigns are still not able to meet the requirements for the investigation of global dust impact on climate, ocean/land geochemistry and ecosystems. Therefore, spaceborne lidars that are capable of observing aerosol have become effective instruments and are widely used in terms of dust plume measurements. The satellite-based lidar CALIOP (Cloud-Aerosol Lidar with Orthogonal Polarization) carried by the platform of CALIPSO (Cloud-Aerosol Lidar and Infrared Pathfinder Satellite Observations) provides us the backscatter coefficient and extinction coefficient at the wavelengths of 532 nm and 1064 nm (Winker et al., 2009). Additionally, the CALIOP product Vertical





Feature Mask product (VFM) presents the aerosol sub-types classification so that the global dust events could be marked. Moreover, large efforts are still needed to monitor the dust emission, transport, dispersion and deposition, and to explore the

dust's impact on the Earth's radiation, climate and ecosystems. Hence, the vertical profiling of the global wind field is necessary to calculate the dust advection. Thanks to the efforts of the European Space Agency (ESA), a first ever spaceborne direct detection wind lidar, Aeolus, which is capable of providing vertical wind fields globally with high temporal and spatial resolution has been developed under the framework of the Atmospheric Dynamics Mission (ADM) (Stoffelen et al., 2005; ESA, 1999; Reitebuch et al., 2012; Kanitz et al., 2019). On 22 August 2018, Aeolus was successfully launched into its sun-

synchronous orbit at a height of 320 km (Witschas et al., 2020; Lux et al., 2020). A quasi-global coverage is achieved daily (~ 15 orbits per day) and the orbit repeat cycle is 7 days (111 orbits). The orbit is sun-synchronous with a local equatorial crossing-time of ~ 6 am/pm. The Atmospheric Laser Doppler Instrument (ALADIN) is a direct detection high spectral resolution wind lidar carried by Aeolus and provides the vertical profiles of the Line-of-Sight (LOS) wind speeds. In order to retrieve the LOS wind speeds, the Doppler shifts of light caused by the emotion of molecules and aerosol particles need to be

identified. Aiming at this, a Fizeau interferometer is applied in the Mie channel to extract the frequency shift of the narrow-band particulate return signal by means of the fringe imaging technique (Mckay, 2002). In the Rayleigh channel, two coupled Fabry-Perot interferometers are used to analyze the frequency shift of the broad-band molecular return signal by the double edge technique (Chanin et al., 1989; Flesia and Korb, 1999).

In the simultaneous observations of the dust plume, the aerosol optical properties can be obtained by means of ALADIN

and CALIOP. By further using the wind vector data from ALADIN, the wind field and relative humidity (RH) data from ECMWF and the trajectories from the HYSPLIT model, the dust transport route can be observed, and the dust advection can be calculated. The paper is organized as follows: in Section 2 the satellite-based instruments, ECMWF and HYSPLIT models are introduced. Section 3 presents the details to the joint dust measurement strategy and methodology. In section 4 we provide the process of the dust event identification and verification as well as the observation results and the dust

advection calculations of the dust transport measurements on 19 June 2020 and during the whole lifetime of the dust event.

## 2. Spaceborne instruments and meteorological models

### 2.1 ALADIN/Aeolus

ALADIN, which is the unique payload of Aeolus, is a direct detection high spectral resolution wind lidar. It is a pulsed ultraviolet lidar working at the wavelength of 354.8 nm with a laser pulse energy around 65 mJ and with a repetition of 50.5

Hz. As the receiver, a 1.5 m diameter telescope collects the backscattered light. The high spectral resolution design of ALADIN allows for the simultaneous detection of the molecular (Rayleigh) and particle (Mie) backscattered signals in two separate channels, each sampling the wind in 24 vertical height bins with a vertical range resolution between 0.25 km and 2.0 km. This makes it possible to deliver winds both in clear and (partly) cloudy conditions down to optically thick clouds at the same time. The horizontal resolution of the wind observations is about 90 km for the Rayleigh channel and about 10 km





for the Mie channel. The detailed descriptions of the instrument design and a demonstration of the measurement concept are introduced in e.g. Reitebuch et al. (2009, 2012), Straume et al. (2018), ESA (2008), Marksteiner et al. (2013), Kanitz et al. (2019), Witschas et al. (2020) and Lux et al. (2020).

The data products of Aeolus are processed at different levels, namely Level 0 (instrument housekeeping data), Level 1B (engineering-corrected HLOS winds), Level 2A (aerosol and cloud layer optical properties), Level 2B (meteorologically-
representative HLOS winds) and Level 2C (Aeolus-assisted wind vectors) (Flament et al., 2008; Tan et al., 2008, 2017). Within the Level 2B processor, the Rayleigh-clear and Mie-cloudy winds are classified, and the temperature and pressure corrections are applied for the Rayleigh wind retrieval (Witschas et al., 2020). In this study, the Level 2A (baseline 10 referring to the L2A processor v3.10) aerosol optical properties and Level 2C (baseline 10 referring to the L2A processor v3.10) wind vectors are used. For the calculation of particle volume concentration distribution and mass concentration, the
extinction coefficients at the wavelength of 355 nm are used.

## 2.2 CALIOP/CALIPSO

Launched in 2006, CALIPSO provides aerosol and cloud optical properties information, e.g., particle depolarization ratio, extinction coefficient, backscatter coefficient and Vertical Feature Mask (VFM) (Winker et al., 2009). The VFM product describes the vertical and horizontal distribution of cloud and aerosol types along the observation tracks of CALIPSO. In this
study, the backscatter coefficients at the wavelengths of 532 nm and 1064 nm from the CALIPSO L2 product are used for the calculation of the dust volume concentration distribution and mass concentration. The VFMs from CALIPSO are also applied to identify the subtypes of aerosol layers. The extinctions from the CALIPSO L2 product are not used in this study, because global average lidar ratio taken for the CALIPSO retrieval is lower than the lidar ratio for Western Saharan dust. The extinctions at 532 nm and 1064 nm used in this study are calculated by the CALIPSO retrieved backscatters and the
corrected lidar ratios: 58 sr at 532 nm (Amiridis et al., 2013), 60 sr at 1064 nm (Tesche et al., 2009).

## 2.3 ECMWF climate reanalysis

Supported by the Copernicus Climate Change Service (C3S), ECMWF provides the atmospheric reanalysis ERA5 which presents a detailed record of the global atmosphere, land surface and ocean waves from 1950 onwards (Hersbach et al., 2020). The 4D-Var assimilated ERA5 produces the hourly vertical profiles (at 37 pressure levels) of global wind fields with
a grid resolution of 31 km. After the successful launch of the Aeolus, the ECMWF started to simulate the wind products of Aeolus from January of 2020. In this study, the wind field data from ECMWF is applied in filling in the missing data within the region between the tracks of Aeolus and CALIPSO and to illustrate the homogeneity of the wind field in this region.



### 2.4 HYSPLIT

The Hybrid Single-Particle Lagrangian Integrated Trajectory model (HYSPLIT) is a modelling system for determining the trajectories, transport and dispersion of air masses developed by the National Oceanic and Atmospheric Administration (NOAA) Air Resources Laboratory (ARL) (Draxler and Hess 1998; Draxler and Rolph 2012). Backward and forward trajectories are the mostly commonly-used model applications to determine the origin of air masses (Stein et al., 2015). In this study, HYSPLIT is used to describe and check the routes of transport, dispersion, and deposition of dust plumes.

### 3. Methodology

In the study of dust transport and advection, as shown in Fig. 1, the dust identification, Aeolus and CALIPSO tracks match, data analysis and HYSPLIT model analysis are described in the schematic flowchart.

### 3.1 Method used to match CALIPSO and Aeolus data

To identify the dust events and to choose the quasi-synchronized observations with ALADIN and CALIOP, the "Dust score index" data provided by AIRS/Aqua are used to determine the dust plume coverage and transport route. With this information, the VFM products from the simultaneous observations with CALIOP are applied to cross-check the identification of dust events. Hence the vertical distributions of dust plumes are obtained. To find the original sources and to predict the transport routes of dust plumes, the backward trajectory and forward trajectory are used respectively. When the dust events are determined, the simultaneous observations with ALADIN and CALIOP have to be selected. Starting from the CALIOP observations, the nearest Aeolus footprints were found. Since the orbits of Aeolus and CALIPSO are different, they cannot meet each other at the exactly same time and same location. From our study, the closest CALIPSO scanning tracks to those of Aeolus, are about 4 hours ahead of Aeolus. Based on the transport directions of dust events modelled with HYSPLIT, the tracks of Aeolus should always be downwind of the tracks of CALIPSO. When the tracks of Aeolus and CALIPSO are selected, the distances between the tracks can be calculated. Assuming the wind speed between CALIPSO scanning tracks and Aeolus is in the range of 5 $m \cdot s^{-1}$ to 15 $m \cdot s^{-1}$, the transport distances of the dust plumes are in the range of 72 km to 216 km. During this short time, dust optical properties remain almost unchanged (Haarig et al., 2017). Consequently, if the distances between two satellites scanning tracks are less than 200 km and the tracks of Aeolus are downwind of the tracks of CALIPSO, it is reasonable to state that the dust plumes captured by CALIPSO are transported towards the Aeolus scanning regions in around 4 hours, hence the following procedures could be continued. To conclude, a successful match at least meets two criteria including 1) the tracks of Aeolus are downwind of the tracks of CALIPSO and 2) the distances between two satellites scanning tracks are less than 200 km.



## 3.2 Datasets and quality control

This study uses the extinction coefficient at 355 nm from ALADIN and the backscatter coefficients at 532 nm and 1064 nm from CALIOP. The extinction coefficient at 355 nm corresponds to the "Aeolus Level 2A Product" retrieved by SCA (standard correction algorithm). In this study, we choose SCA instead of ICA (iterative correction algorithm) because the extinction coefficients from ICA are noisy and the assumption of "one single particle layer filling the entire range bin" in SCA is met in the situation of the heavy dust events. Additionally, we use the mid bin product (sca_optical_properties_mid_bins) of SCA instead of the normal product of SCA, because the mid-bin algorithm provides more robust results (Baars et al., 2021; Flament et al., 2021). The extinction coefficient, which is more sensitive to noise and is the significant input of the dust advection calculation, is better retrieved through this "mid bin" averaged version of the algorithm. In terms of quality control, negative extinction coefficient values of L2A are excluded while the "bin_1_clear" flag and the "processing_qc_flag" of L2A are used to eliminate invalid data. The backscatter coefficients at 532 nm and 1064 nm are the "Total_Backscatter_Coefficient_532" and "Backscatter_Coefficient_1064" from CALIPSO. Since the footprints of Aeolus and CALIPSO are not exactly matched, the missing wind data between their tracks have to be filled in using the ERA5 wind field data. There are two reasons for using the ERA5 wind field data between Aeolus and CALIPSO tracks. One is that the ERA5 wind speed and direction data provide the evidence of dust transport from CALIPSO tracks towards Aeolus tracks. Secondly, the ERA5 wind field data between the tracks of Aeolus and CALIPSO at all height surfaces are smoothly distributed and the values are stable. However, the Aeolus L2C data can be used at the location of the CALIPSO track.

## 3.3 Dust advection calculation

In Fig. 2, the flowchart of dust mass advection calculation procedure is provided. Based on the dataset consists of the backscatter coefficients and extinction coefficients at the wavelengths of 1064 nm and 532 nm from CALIOP and the extinction coefficients at the wavelength of 355 nm from ALADIN, the aerosol volume concentration distribution can be calculated based on the regularization method which was performed by generalized cross-validation (GCV) from Müller et al. (1999). The lidar ratio for Western Saharan dust is higher than the global average taken for the CALIPSO retrieval, thus the extinctions from the CALIPSO L2 product are not used in the calculation of the aerosol volume concentration. The extinctions at 532 nm and 1064 nm utilized for the regularization method are calculated from the CALIPSO backscatter and the corrected lidar ratios: 58 sr at 532 nm (Amiridis et al., 2013), 60 sr at 1064 nm (Tesche et al., 2009).

The advantage of this method is that it does not require prior knowledge of the shape of the particle size distribution and the estimated uncertainty of aerosol volume concentration is on the order of 50% if the estimated errors of the inputs are on the order of 20%. For the backscatter coefficient at 532 nm, during the daytime, the average difference between collocated CALIPSO and HSRL measurements is 1.0%±3.5 % in V4 (Getzewich et al., 2018); for the backscatter coefficient at 1064 nm, the CALIOP V4 1064 nm calibration coefficients are accurate to within 3 % (Vaughan et al., 2019). Consequently, we



consider that the uncertainties of CALIPSO-retrieved extinction and backscatter coefficients to be of order 20%. According to Flament et al. (2021), because of the lack of cross-polarized light, backscatter coefficients at 355nm of Aeolus are

185 underestimated, especially for dust aerosol. Nevertheless, the extinction is not affected. In this work, Aeolus retrieved backscatter coefficients at 355nm are not applied for the calculations of the dust volume concentration distribution and mass concentration. For the accuracy of the Aeolus-retrieved extinction coefficient, the simulation extinction coefficients fit the inputs well mostly, especially when the altitude is larger than 2 km (Flament et al, 2021). Hence, we consider that after rigorous quality control, the Aeolus L2A extinction coefficient could be the input parameters of the regularization method. In

conclusion, the estimated errors of the five input parameters we used to calculate the aerosol volume concentration are on the order of 20%. The estimate errors of dust advection are the combination of mass concentration estimate errors (~50%) and Aeolus L2C wind vector estimate errors.

It should be emphasized that due to the different vertical and horizontal resolution between Aeolus and CALIPSO data, a common pixel grid should be established before calculation. For vertical resolution, 23 data bins of Aeolus L2A mid bin

optical property products are interpolated to 399 data bins of CALIPSO according to the altitude information of two products. For horizontal resolution, both Aeolus and CALIPSO products are averaged along every integer latitude to acquire a common horizontal pixel grid. After integrating and multiplying an assuming typical dust particle density which is set as 2.65 $g \cdot cm^{-3}$ referring to previous studies (e.g., Schepanski et al., 2009; Hofer et al., 2017; Mamouri and Ansmann, 2017), the particle mass concentration is estimated following the method of Engelmann et al. (2008). ECMWF wind field data and

RH data between Aeolus and CALIPSO scanning tracks are averaged along longitude and averaged along every integer latitude, while, vertically, they are interpolated to CALIPSO data bins to match the common pixel grid. Since the observations with ALADIN and CALIOP are not exactly simultaneous, the ECMWF wind field data between the two spaceborne lidars' scanning tracks is utilized to illustrate the homogeneity of the wind field between two tracks, so that the Aeolus L2C wind vector data along the Aeolus tracks can represent the wind field of the whole area and can be employed in

the calculation of the dust mass advection. In the transport regions of the dust plume (between 5 °N and 30 °N), if both of the standard deviation percentages of wind speed and direction along each latitude line are less than 10%, it is considered that the wind fields between the two spaceborne lidars' scanning tracks are homogeneous and stable. Besides, when the RH is larger than 90%, the dust aerosol will be influenced by the hygroscopicity effect and its properties could change. Then the mass concentration calculation method does not make sense any more (Engelmann et al., 2008). For the cloud screening,

aside the RH data, we use Level 2 5 km aerosol profile of CALIPSO, which only provide aerosol optical properties so the cloud can be screened. Therefore, relative humidity data provided by ECMWF is used to filter unavailable data. Ultimately, combining the particle mass concentration and the horizontal wind vector provided by Aeolus L2C product, the dust mass advection is defined as Eq. (1), to represent the transportation of dust aerosol quantitatively.

$$\overrightarrow{Advection}_{aerosol-mass} = m \cdot \vec{v}, \tag{1}$$





where $m$ is the aerosol mass concentration and $\vec{v}$ is the horizontal wind vector.



**Figure 1. Dust identification, Aeolus and CALIPSO tracks match and data procedures.**



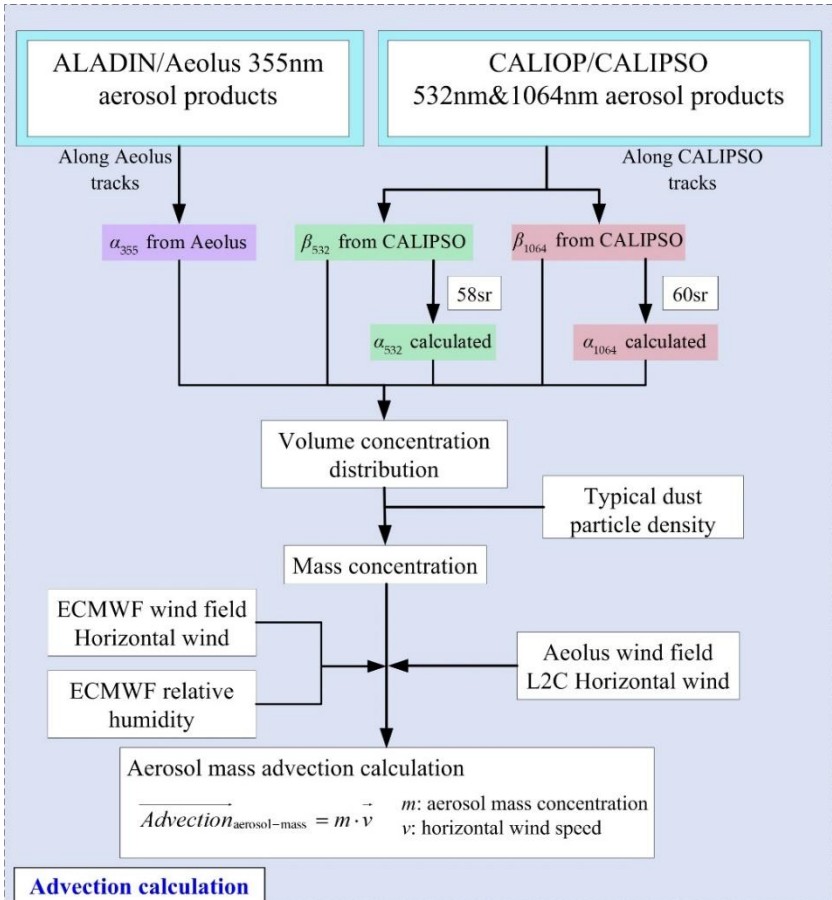

**Figure 2. The flowchart of the dust mass advection calculation procedure.**

## 4. Results and discussion

### 4.1 Dust identification and verification by AIRS/Aqua, CALIOP and HYSPLIT

During 14 and 27 June 2020, a complete dust event process, including dust emission, transportation, dispersion and deposition, took place in the regions of Africa, Atlantic Ocean and the Americas. In Fig. 3, the "Dust Score Index" provided by AIRS/Aqua at different stages are presented. From this figure, the long-term dust event generated on 14 and 15 June 2020 from the Sahara Desert in North Africa dispersed and moved westward over the Atlantic Ocean, finally being deposited in the western part of the Atlantic Ocean, the Americas and the Caribbean Sea. It should be emphasized that since the dust scores are provided per day, the dust events are just preliminarily classified. From the spaceborne CALIOP observations, it is found that sometimes dust events are actually present but are misjudged by AIRS/Aqua, which may result from interference from high-altitude cloud layers. The daily dust score data over the Sahara-Atlantic-Americas region generally reveals the transportation of the dust plume horizontally.





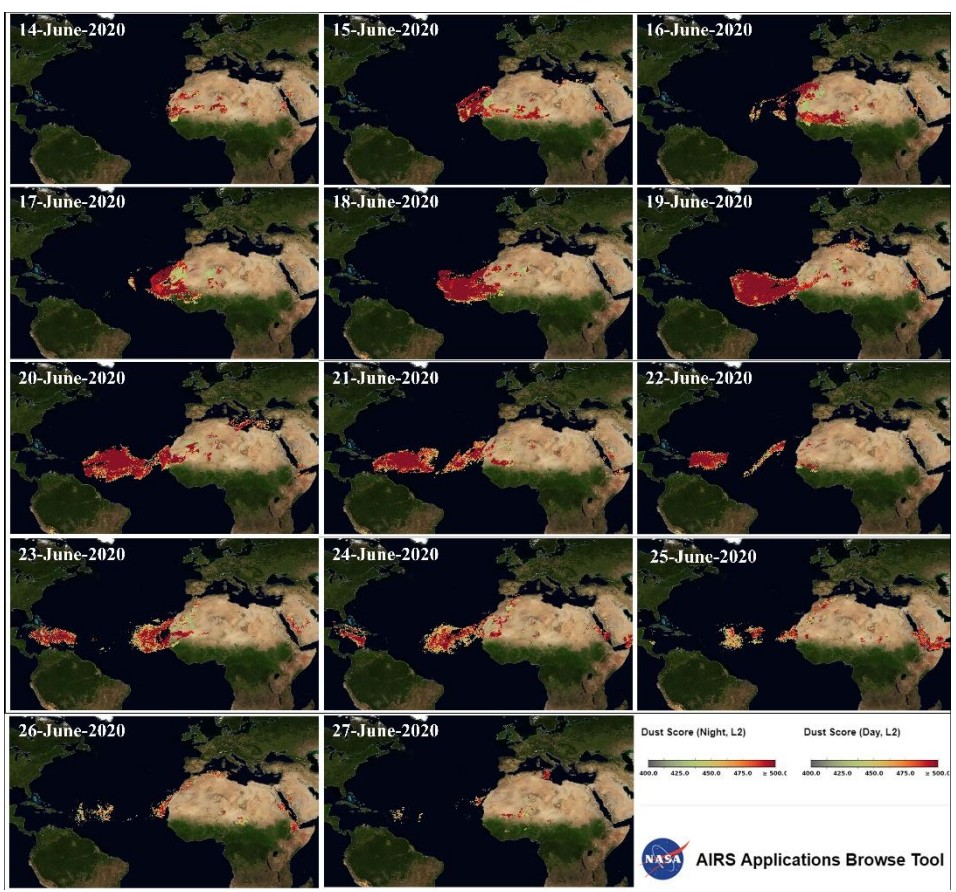

**Figure 3. The Dust Score Index provided by AIRS/Aqua at different stages, including emission, transportation, dispersion and deposition (https://airs.jpl.nasa.gov/map/, last access: 10 January 2022).**

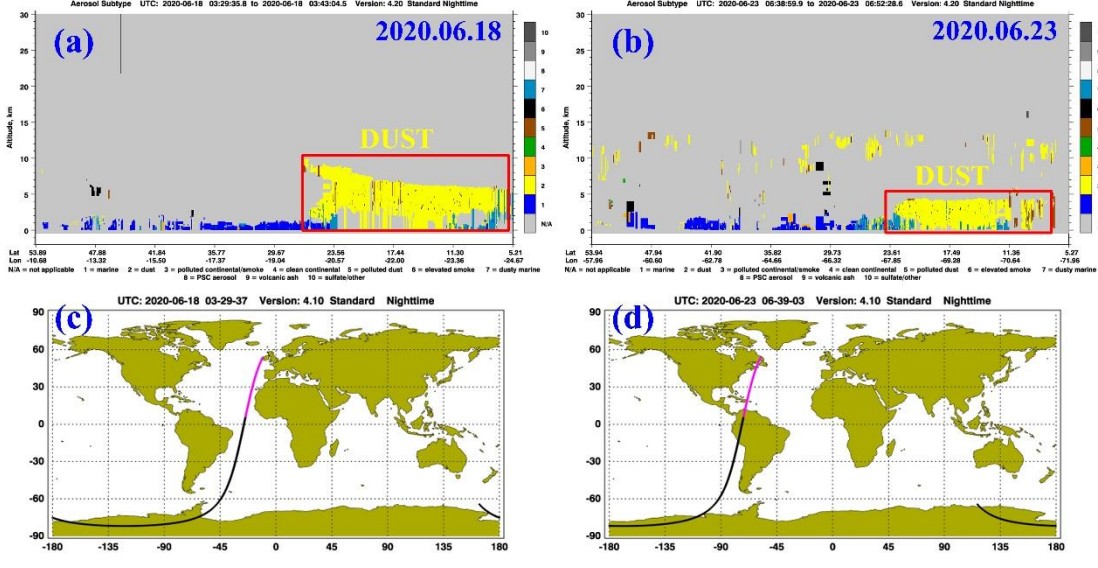





**Figure 4. VFM from CALIPSO L2 product (a) on 18 June 2020 over the eastern Atlantic (around west coast of Sahara) and (b) on 23 June 2020 over the western Atlantic (around east coast of America). (c) and (d) show the corresponding CALIOP scanning tracks of (a) and (b) (https://www-calipso.larc.nasa.gov/products/lidar/browse_images/production/, last access: 10 January 2022).**

Figure 4 presents the vertical distribution of the dust plume during the development phase (18 June 2020) over the eastern Atlantic and during the deposition phase (23 June 2020) over the western Atlantic. From Fig. 4 (a), it can be seen that the dust plume has been lifted up to around 10 km. Figure 4 (b) presents the descending dust plume, the bottom of which may mix with marine aerosol and become dusty marine aerosol. Therefore, the VFM data of CALIPSO captures the dust plume vertically over the eastern and the western Atlantic and verifies the dust transportation process.





**Figure 5. (a)(c)(e) CALIPSO total backscatter coefficient profiles and particle depolarization ratio profiles capturing dust layers at around 0500UTC 20 June 2020. (b)(d)(f) HYSPLIT backward trajectories and forward trajectories at different sites of corresponding CALIPSO profiles and different heights on 0500UTC 20 June 2020. The backward and forward trajectories' durations are both 144 hours (https://www.ready.noaa.gov/hypub-bin/trajtype.pl?runtype=archive, last access: 10 January 2022).**

To cross-check the transport route of the dust events, three adjacent CALIPSO profiles from one orbit capturing a dust aerosol layer are shown in Fig. 5. Meanwhile, the backward and forward trajectories of these three sources starting at 0500UTC 20 June 2020 with the NOAA HYSPLIT model were conducted. CALIPSO total backscatter coefficient profiles and particle depolarization ratio profiles of source A ( 23.60°N,39.48°W ), source B ( 17.47°N,40.91°W ) and source C ( 11.34°N,42.27°W ) are shown in Fig. 5 (a), (c) and (e). It can be seen that the altitude range of dust plume layers at source

A and source B are approximately 3 km to 6 km, while the dust layer at source C is lower, is around 2 km to 4km. From Fig. 5 (b), (d) and (f), it is seen that the dust plumes at source A and source B are mainly generated from the centre of the Sahara Desert while the dust plumes at source C at about 3 km and 2 km originated from the west side of the Sahara Desert. The forward trajectories clearly indicate that the dust plumes were separated into two directions. The dust plumes transportation directions of source A, B and C are toward Central/South America respectively and the Caribbean Sea. At the end of these

three forward trajectories, the altitudes of the dust aerosol reduce to around 1km, which indicates a descent of dust plumes. After 26 June, transported over the whole Atlantic Ocean, most of the dust plumes were settled in the western Atlantic Ocean, Central America and the Caribbean Sea.



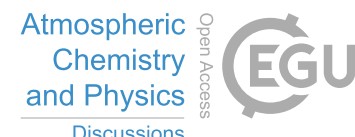

## 4.2 Measurement case and dust advection calculation on 19 June 2020

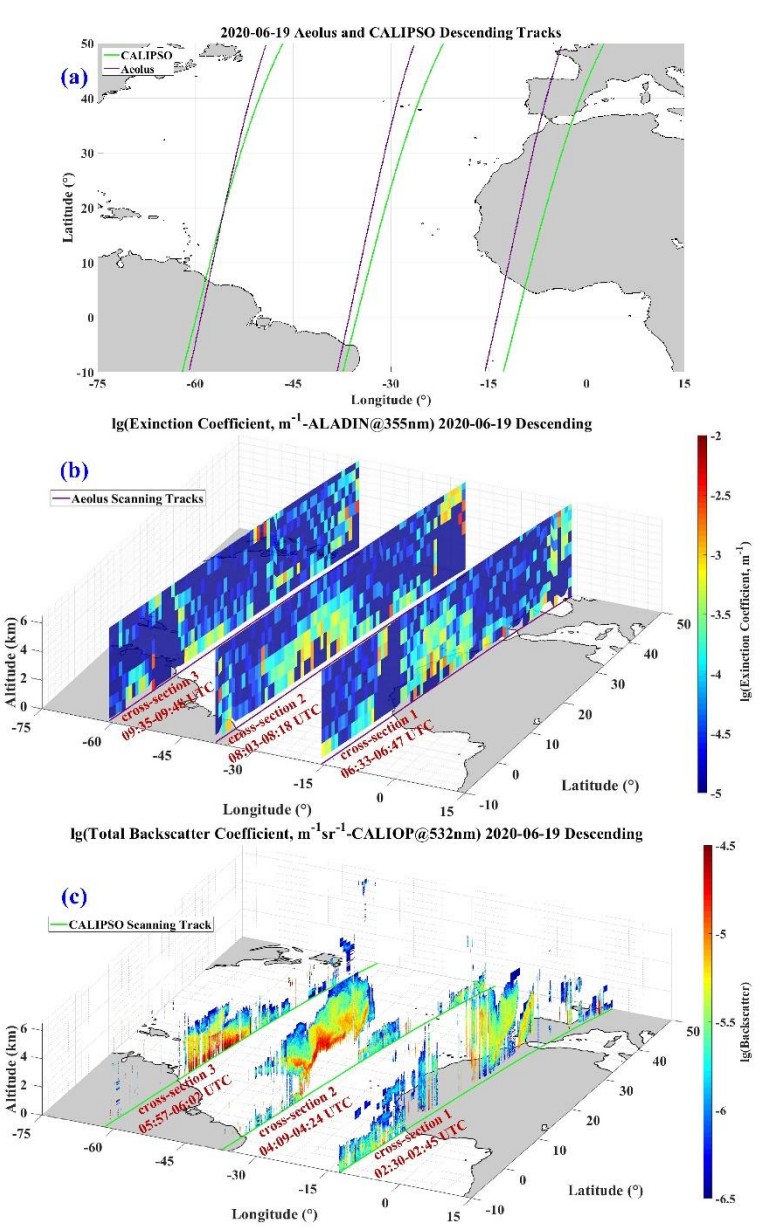

**Figure 6. Observation tracks of Aeolus and CALIPSO on 19 June 2020. The purple lines indicate the tracks of Aeolus and the green lines indicate the tracks of CALIPSO. (a) Vertical view of Aeolus and CALIOP tracks and HYSPLIT trajectories; (b) Extinction coefficient cross-sections measured with Aeolus and (c) Total backscatter coefficient cross-sections measured with CALIOP.**

In this section, the dust event measurement case that occurred on 19 June 2020 is introduced in detail. The quasi-

synchronized observations from ALADIN and CALIOP on 19 June 2020 are presented in Fig. 6, where the purple lines indicate the tracks of Aeolus and the green lines indicate the tracks of CALIPSO. From the profiling of dust optical





properties, discriminated by the CALIOP measurements, the dust dispersion over Atlantic Ocean on this day could be determined. The extinction coefficients and backscatter coefficients at the wavelengths of 355 nm, 532 nm and 1064 nm within the dust mass are also determined. From the profiling, it was found that the mean backscatter coefficients at 532 nm

were about $3.88\times10^{-6} \pm 2.59\times10^{-6}$ m$^{-1}$sr$^{-1}$ in "cross-section 1", $7.09\times10^{-6} \pm 3.34\times10^{-6}$ m$^{-1}$sr$^{-1}$ in "cross-section 2" and

$7.76\times10^{-6} \pm 3.74\times10^{-6}$ m$^{-1}$sr$^{-1}$ in "cross-section 3". On 19 June 2020, the main part of the dust plume was transported in the free troposphere over the eastern Atlantic, which can be verified in Fig. 3. "Cross-section 2" is in the middle of the dust plume, while "cross-section 1" captured the dust layer over the western Sahara, which is the emission region of this dust event. However, the emission intensity of the dust from the emission region became weak on 19 June 2020. Thus, the

backscatter coefficients in "cross-section 2" are larger than those of "cross-section 1". Besides, the backscatter coefficients of the dust layer in "cross-section 3" slightly increased which may result from the fact of its mixture with other aerosol types (e.g., marine aerosol) near the ocean surface.

Based on the extinction coefficient at 355nm, the backscatter coefficients and extinction coefficients at 532nm and 1064nm, combined with the wind vector data from ALADIN, the dust advection can be calculated. The L2C wind product

provided by Aeolus results from the background assimilation of the Aeolus HLOS winds in the ECMWF operational prediction model. The u and v components of the wind vector and supplementary geophysical parameters are contained in L2C data product. From literature reports (e.g., Lux et al., 2020), the Aeolus L2B Rayleigh LOS winds and the ECMWF model LOS winds show good agreement with a correlation coefficient of 0.92 and mean bias of 1.62 $m \cdot s^{-1}$. As introduced in Section 3.3, the ECMWF wind field data between the two spaceborne lidars scanning tracks is utilized to illustrate the

homogeneity of the wind field between two tracks. Hence, in this study, if the wind fields between tracks is stable, the "analysis_zonal_wind_velocity" and "analysis_meridional_wind_velocity" from the Aeolus L2C wind vector product could be applied for the calculation of the dust advection.

To calculated the dust advection during this event, the wind field and relative humidity information are necessary. Since the observations with ALADIN and CALIOP are not exactly simultaneous, the stability of the wind field between the

scanning tracks of them has to be estimated. Hence, the wind speed, wind direction and relative humidity between the tracks are analysed with the data from ECMWF, as presented in Fig. 7. From this figure, the wind speed, wind direction and relative humidity at the height surfaces of 1 km, 2 km, 3 km, 4 km, 5 km and 6 km are shown as examples. The wind fields between the tracks of Aeolus and CALIPSO at all height surfaces are smoothly distributed and the values are stable. Thus, the wind vector data from Aeolus L2C could be applied in the calculation of dust advection. It should be emphasized that,

during the calculations of the dust advection, the results with relative humidity higher than 90% have to be removed.

In order to verify the retrieval results of the regularization method, we compare the mass concentration retrieved by the regularization method (the retrieval method) with the results calculated directly using the mass-specific extinction coefficient (the factor method) (Ansmann et al., 2012). According to Ansmann et al. (2012), the aerosol mass concentration can also be calculated by the method that the extinction coefficient at 532 nm divides the mass-specific extinction coefficient. Hence, the



reference mass concentration of every cross-sections is calculated with the CALIPSO extinction coefficient at 532 nm along tracks and the Sahara dust mass-specific extinction coefficient ( $0.52\,\mathrm{m}^2\cdot\mathrm{g}^{-1}$ ). Table 1 shows the mean mass concentration of every cross-section on 19 June 2020 calculated by the two methods. Referring to Ansmann et al. (2012), Ansmann et al. (2017) and Haarig et al. (2019), the mass concentrations of typical dust layers from Sahara vary from 0.05 $\mathrm{mg}\cdot\mathrm{m}^{-3}$ to 0.5 $\mathrm{mg}\cdot\mathrm{m}^{-3}$. Moreover, this dust event is a historic and massive Saharan dust intrusion into the Caribbean Basin and southern

US, which is nicknamed the "Godzilla" by Yu et al. (2021). Thus, it should be introduced that in the process of average calculation, the mass concentration values smaller than 0.05 $\mathrm{mg}\cdot\mathrm{m}^{-3}$ or larger than 0.5 $\mathrm{mg}\cdot\mathrm{m}^{-3}$, which are unreasonable, are excluded. From the comparison, it can be found that the results from the factor method are larger than the results from the retrieval method. However, considering the errors of these two methods, we consider that the mass concentration estimated by the regularization method is reasonable and acceptable.

**Table 1. Mean dust mass concentration of each cross-sections on 19 June 2020 calculated by two methods**

| Cross-section | 1 | 2 | 3 |
|---|---|---|---|
| Mean mass concentration, $\mathrm{mg}\cdot\mathrm{m}^{-3}$ (the retrieval method) | 0.19 | 0.17 | 0.15 |
| Mean mass concentration, $\mathrm{mg}\cdot\mathrm{m}^{-3}$ (the factor method) | 0.24 | 0.26 | 0.20 |

In Fig. 8, the dust advection at different heights of the three cross-sections are presented. It can be seen that, dominated by the Northeasterly Trade-wind between the latitudes of $5\,°\mathrm{N}$ and $30\,°\mathrm{N}$, the dust plumes are mainly transported to the west part of the Atlantic Ocean. From the profiling, the mean dust advection value is about $2.06\,\mathrm{mg}\cdot\mathrm{m}^{-2}\cdot\mathrm{s}^{-1}$ in "cross-section 1" (dust portion during emission phase), $1.47\,\mathrm{mg}\cdot\mathrm{m}^{-2}\cdot\mathrm{s}^{-1}$ in "cross-section 2" (dust portion during development phase) and

$0.95\,\mathrm{mg}\cdot\mathrm{m}^{-2}\cdot\mathrm{s}^{-1}$ in "cross-section 3" (dust portion during deposition phase), respectively. The lowest value of the mean dust advection appears in "cross-section 3", perhaps because cross-section 3 is farthest from the source region. During the dispersion and deposition processes of dust aerosol transportation, it is reasonable that the lowest value appears in cross-section 3.





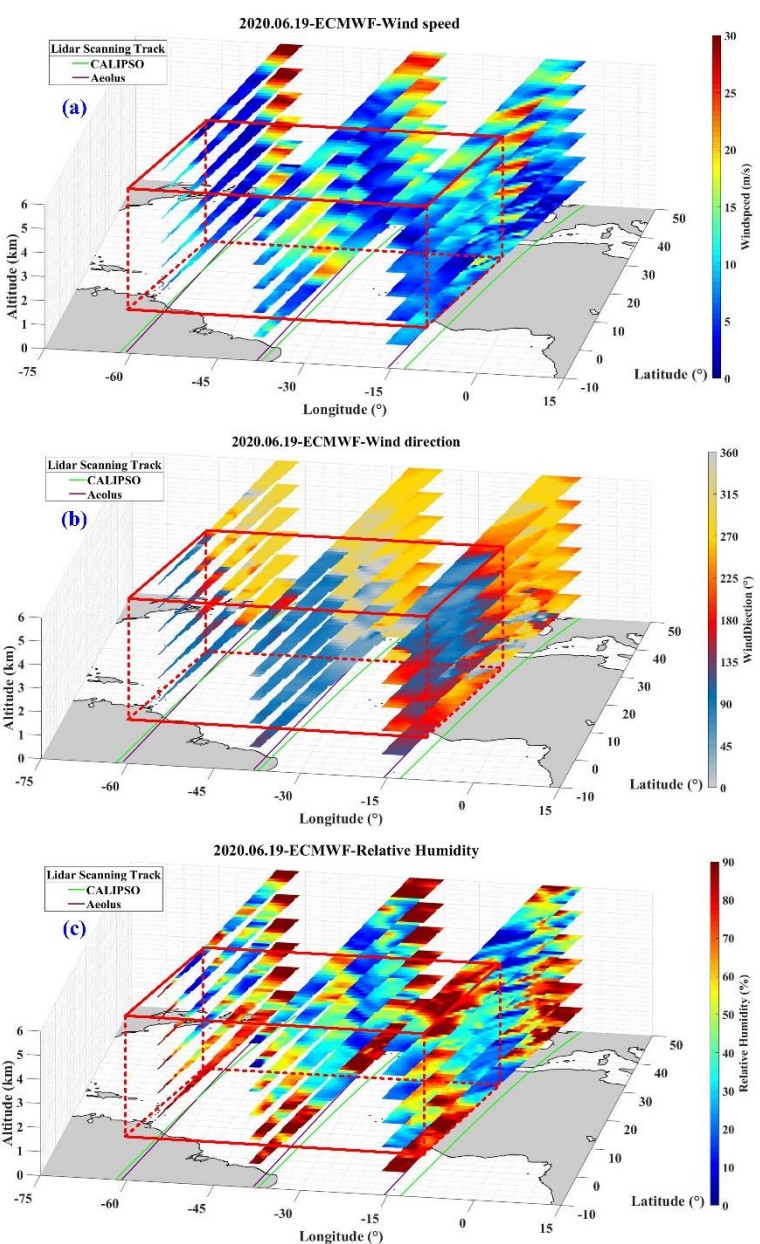

**Figure 7. The wind speed, wind directions and relative humidity between the quasi-synchronization observation tracks of Aeolus and CALIPSO provided by ECMWF on 19 June 2020. The red frames indicate the transport region of the dust plume.**





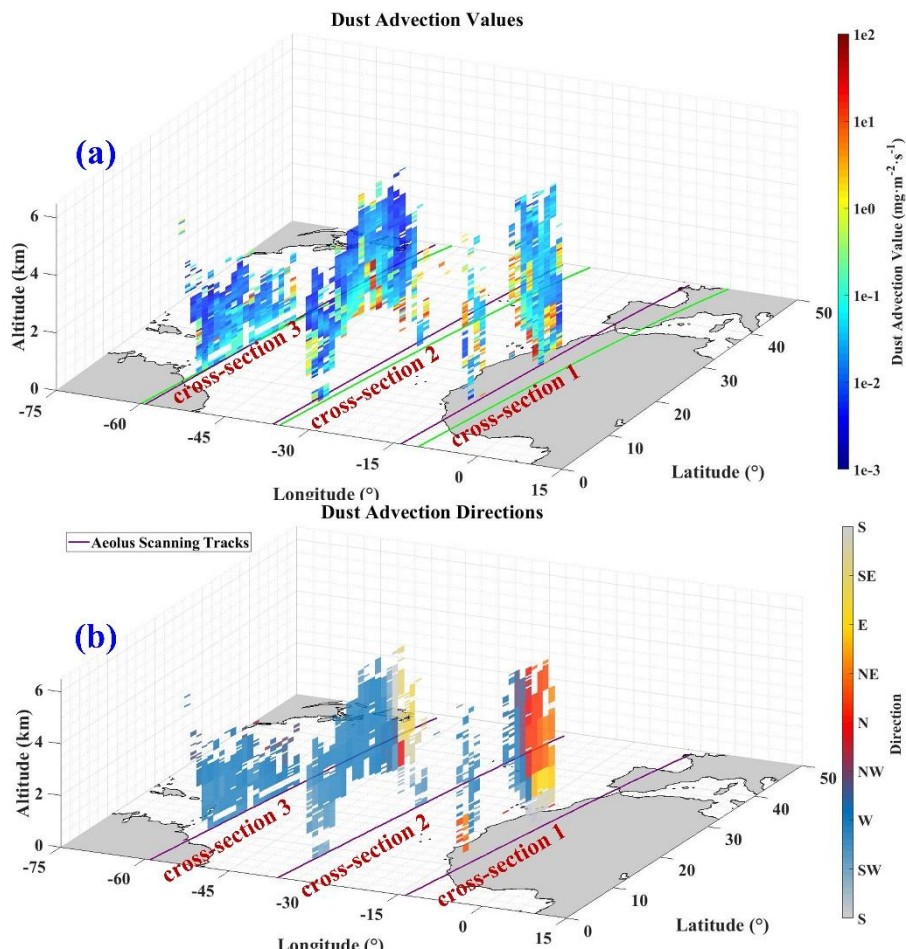

**Figure 8. The dust advection calculated with data from ALADIN, CALIOP and ECMWF (a) the dust advection**
**values at different cross-sections of dust plumes and (b) the dust advection directions at different cross-sections of**
**dust plumes on 19 June 2020.**

**4.3 Dust advection during the lifetime of dust event during 14 June and 27 June 2020**

During this dust event, the quasi-synchronized observations with ALADIN and CALIOP were selected to follow the
transport and dispersion of dust. The detailed information about the ALADIN and the CALIOP observations on 15, 16, 19,
24, 27 June 2020 along the transport route and the HYSPLIT modelling are shown in Fig. 9. In Fig. 9(a), the tracks of
ALADIN and CALIOP on those days are indicated by dark purple lines and green lines, respectively. Additionally, the
forward trajectories starting from 19 June and backward trajectories ending at 19 June are modelled and presented in dark
red lines and light purple lines, respectively. In Fig. 9(b) and (c), 5 cross-sections of extinction coefficient at 355 nm
measured at different times with Aeolus and 5 cross-sections of backscatter coefficient at 532 nm measured at different times
with CALIOP are plotted, respectively. From these figures, we can find that the dust transport modelled with HYSPLIT
match well with the dust masses at different cross-sections of Aeolus and CALIPSO. In Fig. 9(d), a side view of the



HYSPLIT trajectories is shown. Consistent with the observations from ALADIN and CALIOP in Fig. 9(b) and (c), there is an apparent descent along the transport route of the dust event.

In Fig. 10, the wind speed and directions at certain height surfaces between the tracks of CALIPSO and Aeolus are shown and are smoothly distributed and the values are stable. Consequently, Aeolus L2C wind vector product can be employed in the calculation of the dust advection. The relative humidity is presented as well.

Table 2 presents the two sets of mean mass concentration of each cross-sections at different times during the dust transport calculated by the retrieval method and the factor method. Compared with the factor method calculation results, it is considered that the dust mass concentration from the retrieval method is reasonable and acceptable.

**Table 2. Mean dust mass concentration of each cross-sections at different times during the dust transport calculated by two methods**

| Date | 15 June | 16 June | 19 June | 24 June | 27 June |
|---|---|---|---|---|---|
| Mean mass concentration, $mg \cdot m^{-3}$ (retrieval method) | 0.19 | 0.20 | 0.17 | 0.19 | 0.17 |
| Mean mass concentration, $mg \cdot m^{-3}$ (factor method) | 0.23 | 0.27 | 0.26 | 0.30 | 0.25 |



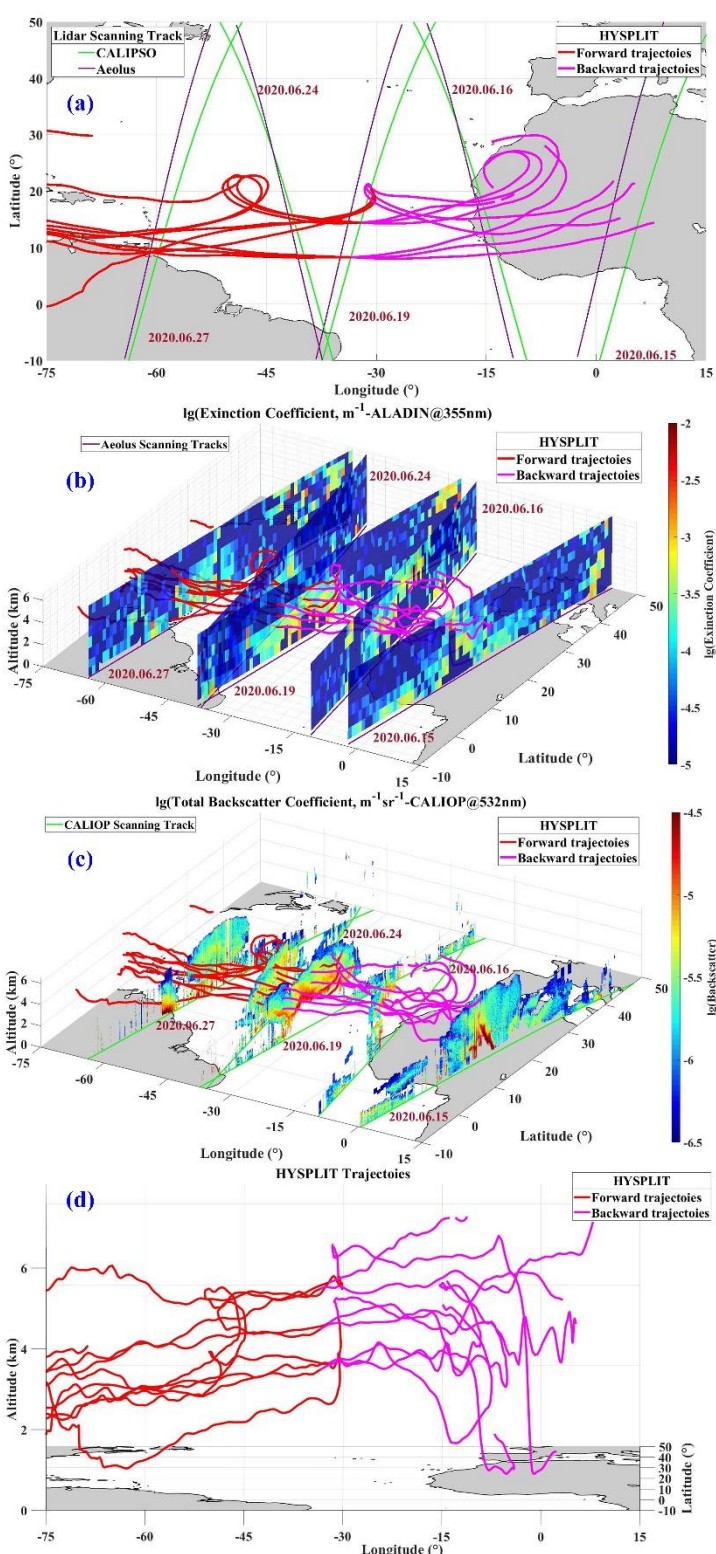





**Figure 9. Observation of dust event during 15 and 27 June 2020 with ALADIN and CALIOP. (a) Vertical view of** 
**Aeolus and CALIPSO tracks and HYSPLIT trajectories; (b) Extinction coefficient cross-sections measured with ALADIN and HYSPLIT trajectories; (c) Total backscatter coefficient cross-sections measured with CALIOP and HYSPLIT trajectories and (d) Side view of HYSPLIT trajectories.**

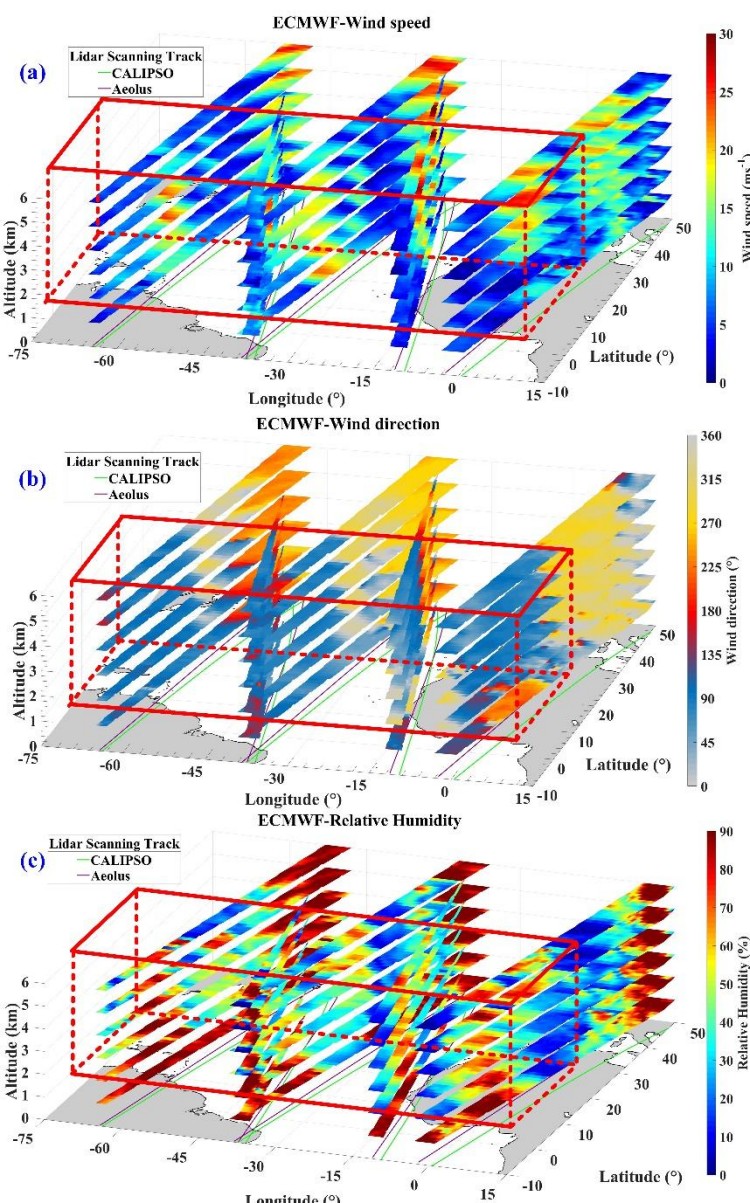

**Figure 10. The wind speed, wind directions and relative humidity between the quasi-synchronization observation** 
**tracks of Aeolus and CALIPSO provided by ECMWF. The red frames indicate the transport region of the dust plume.**





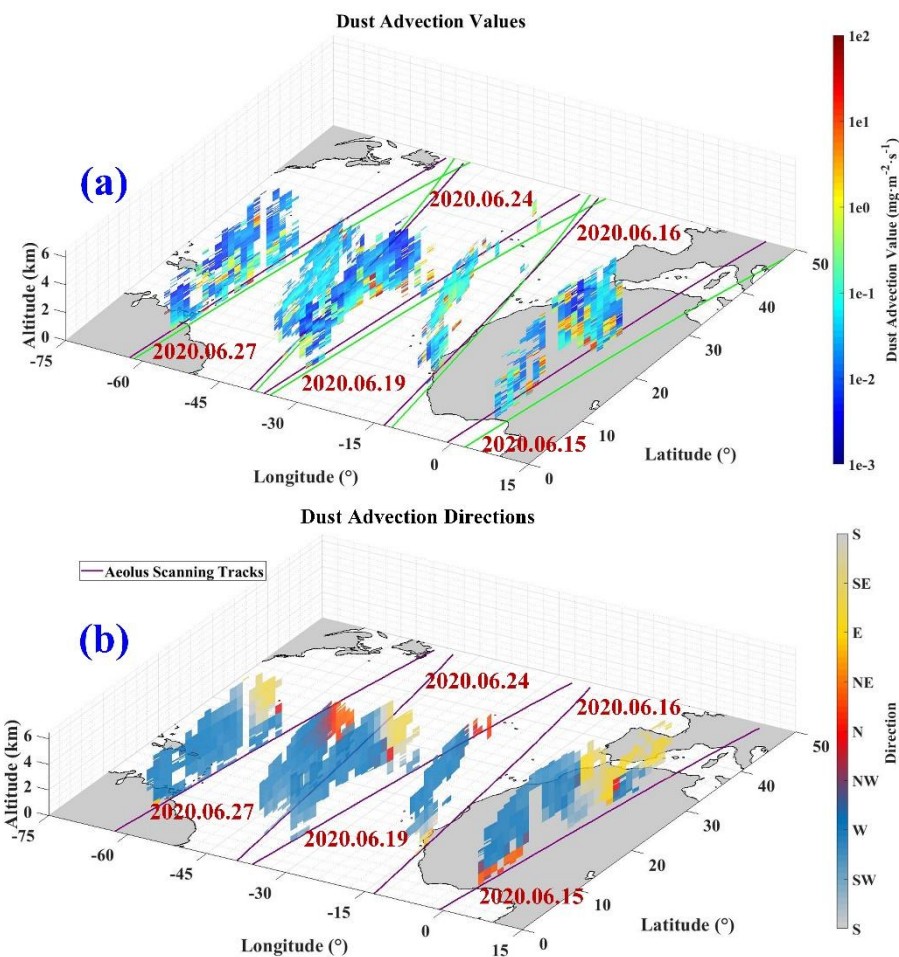

**Figure 11. The dust advection calculated with data from ALADIN, CALIOP and ECMWF (a) dust advection values at different cross-sections and at different times during the dust transport and (b) dust advection directions at different cross-sections and at different times during the dust transport.**

In Fig. 11, the dust advections at different heights of all the cross-sections during the dust transport are presented. In Fig. 11(a), the mean dust mass advection values are about $1.50 \, \mathrm{mg \cdot m^{-2} \cdot s^{-1}}$ on 15 June 2020, $2.41 \, \mathrm{mg \cdot m^{-2} \cdot s^{-1}}$ on 16 June 2020, $1.47 \, \mathrm{mg \cdot m^{-2} \cdot s^{-1}}$ on 19 June 2020, $2.01 \, \mathrm{mg \cdot m^{-2} \cdot s^{-1}}$ on 24 June 2020 and $1.15 \, \mathrm{mg \cdot m^{-2} \cdot s^{-1}}$ on 27 June 2020. From this trend, it appears that the mean advection value (around $1.50 \, \mathrm{mg \cdot m^{-2} \cdot s^{-1}}$) on 15 June when the dust originated is lower than that (around $2.41 \, \mathrm{mg \cdot m^{-2} \cdot s^{-1}}$) on 16 June. It has to be emphasized that, according to Fig. 9(a), Aeolus and CALIPSO quasi-synchronically observed the dust plumes on 15 June only over part (not whole) of the emission regions. The emission part from West Africa is missed and thus leads to the lower mean dust advection value on 15 June than that on 16 June. With the development and enhancement of the dust event, the mean advection value gradually increases and reaches the peak



value (around $2.41\,\mathrm{mg\cdot m^{-2}\cdot s^{-1}}$) on 16 June. Then, during the transport of the dust plume over the Atlantic Ocean, the mean

advections decreased on 19 June and 24 June. Ultimately, resulting from the dispersion and deposition of the dust plume to the west part of Atlantic Ocean, the Americas and the Caribbean Sea, the dust advection on 27 June becomes the lowest ($1.15\,\mathrm{mg\cdot m^{-2}\cdot s^{-1}}$) of the whole dust transportation.

From Fig. 12, the L2C wind vectors including u and v components from Aeolus at different times are plotted. In Figure 12(a), the dust plumes are trapped in the Northeasterly Trade-wind zone (indicated by the blue colour at different cross-

sections) between the latitudes of $5\,°\mathrm{N}$ and $30\,°\mathrm{N}$ and altitudes of 0 km and 6 km. The u component values of the wind vectors in the trade-wind zone are high, reaching 20 $\mathrm{m\cdot s^{-1}}$. Dominated by the trade-wind, the dust plumes are mainly transported to the west. This space area looks like a tunnel and the dust plumes are transported inside. Since the dust plumes are frequently occurred in this area, this tunnel can be called as "Saharan dust westward transport tunnel". From Figure 12(b), the v component values of the wind vectors are presented as well. Effected by the small wind towards south, the dust plumes

are slightly shifted to the south part of Atlantic Ocean in this case.

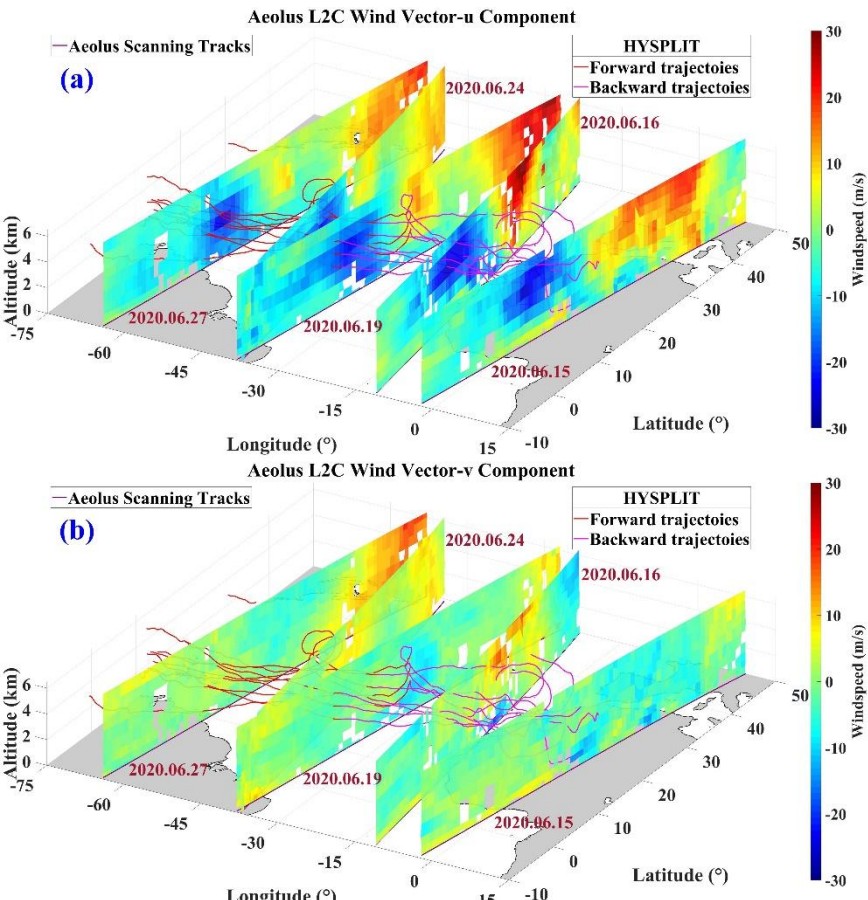

**Figure 12. The u and v components provided by Aeolus and the HYSPLIT model for the dust event.**



## 5. Summary and conclusions

In this study, a long-term large-scale Saharan dust transport event that occurred between 14 June and 27 June 2020 is tracked and its mass advection is calculated with the remote measurement data from ALADIN and CALIOP, the reanalysis data from ECMWF, and HYSPLIT. This allows us to (1) evaluate the performance of the ALADIN and CALIOP on the observations of dust optical properties and wind fields and (2) explore the capability of tracking the dust events and in calculating the dust mass advection.

We identified the dust plumes with AIRS/Aqua Dust Score Index and with the Vertical Feature Mask products from CALIOP. The emission, dispersion, transport and deposition of the dust event are followed using the data from HYSPLIT, CALIOP and AIRS/Aqua. With the quasi-synchronized observations from ALADIN and CALIOP, combined with the wind field and relative humidity from ECMWF, the dust advection is calculated.

From this study, it is found that the dust event generated on 14 and 15 June 2020 from the Sahara Desert in North Africa dispersed and moved westward over the Atlantic Ocean, finally being deposited in the west part of Atlantic Ocean, the Americas and the Caribbean Sea. During the transport and deposition processes, the dust plumes were trapped and transported in the Northeasterly Trade-wind zone between the latitudes of $5\,°\mathrm{N}$ and $30\,°\mathrm{N}$ and altitudes of 0 km and 6 km (we name this space area as "Saharan dust westward transport tunnel"). From the measurement results on 19 June 2020, influenced by the hygroscopic effect and mixing with other types of aerosols, the backscatter coefficients of dust plumes are increasing along the transport routes, with $3.88{\times}10^{-6}\pm2.59{\times}10^{-6}\ \mathrm{m^{-1}sr^{-1}}$ in "dust portion during emission phase", $7.09{\times}10^{-6}\pm3.34{\times}10^{-6}\ \mathrm{m^{-1}sr^{-1}}$ in "dust portion during development phase" and $7.76{\times}10^{-6}\pm3.74{\times}10^{-6}\ \mathrm{m^{-1}sr^{-1}}$ in "dust portion during deposition phase".

Finally, the advection at different dust parts and heights on 19 June and on the entire transport routine during transportation are computed, as shown in Fig. 7 and Fig. 10, respectively. On 19 June, the mean dust advection values are about $2.06\ \mathrm{mg\cdot m^{-2}\cdot s^{-1}}$ during the emission phase, $1.47\ \mathrm{mg\cdot m^{-2}\cdot s^{-1}}$ during the development phase and $0.95\ \mathrm{mg\cdot m^{-2}\cdot s^{-1}}$ during the deposition phase. In the whole life-time of the dust event, the mean dust advection values are about $1.50\ \mathrm{mg\cdot m^{-2}\cdot s^{-1}}$ on 15 June 2020, $2.41\ \mathrm{mg\cdot m^{-2}\cdot s^{-1}}$ on 16 June 2020, $1.47\ \mathrm{mg\cdot m^{-2}\cdot s^{-1}}$ on 19 June 2020, $2.01\ \mathrm{mg\cdot m^{-2}\cdot s^{-1}}$ on 24 June 2020 and $1.15\ \mathrm{mg\cdot m^{-2}\cdot s^{-1}}$ on 27 June 2020. During the dust development stage, the mean advection values gradually increase and reach the maximum value on 16 June with the enhancement of the dust event. Then, the mean advection values decrease since most of the dust was deposited in the Atlantic Ocean, the Americas and the Caribbean Sea.

**Data availability.**

The Aeolus data are downloaded via the website https://aeolus-ds.eo.esa.int/oads/access/collection (last access: 10 January 2022). The Aeolus L2A and L2C data we used in this paper are not available publicly at the time the article was submitted. We are allowed to access the data through our participation as a Calibration and Validation team. The CALIOP data can be
downloaded from https://eosweb.larc.nasa.gov/project/CALIPSO (last access: 10 January 2022). The ECMWF reanalysis ERA5 wind data can be accessed from https://cds.climate.copernicus.eu/cdsapp#!/dataset/reanalysis-era5-pressure-levels?tab=form (last access: 10 January 2022). The backward trajectory and forward trajectory of HYSPLIT can be run at https://www.ready.noaa.gov/HYSPLIT_traj.php (last access: 10 January 2022).

**Author contributions.**

G. Dai and S. Wu conceived of the idea for the dust transport and mass advection measurement with spaceborne lidars ALADIN, CALIOP and model reanalysis data; G. Dai and K. Sun wrote the manuscript; K. Sun, G. Dai, S. Wu, B. Liu and Q. Liu conducted the data analyses; X. Wang helped in programming, X. E downloaded the ECMWF data, and all the co-authors discussed the results and reviewed the manuscript.

**Competing interests.**

The authors declare that they have no conflict of interest.

**Special issue statement.**

This article is part of the special issue "Aeolus data and their application". It is not associated with a conference.

**Acknowledgments.**

This study has been jointly supported by the National Key Research and Development Program of China under grant
2019YFC1408001 and 2019YFC1408002, the National Natural Science Foundation of China (NSFC) under grant 41905022 and 61975191 and the Key Research and Development Program of Shandong Province (International Science and Technology Cooperation) under Grant 2019GHZ023.



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
