# Peer review of "Dust transport and advection measurement with spaceborne lidars ALADIN, CALIOP and model reanalysis data"

_Atmospheric Chemistry and Physics, 2022_

## Author Comment (AC1)

The combination of two space-based lidars (CALIPSO and Aeolus) is new and deserves attention. The Saharan dust transport across the Atlantic Ocean is a well-known large-scale phenomenon and suited to demonstrate the novel approach. Personally, I welcome the resubmission of the now improved version of the manuscript. However, there are still some major reviews necessary till the final publication.

Major comments:

1. The comparison of the 3 cross sections on 19 June 2020 is misleading (Section 4.2). With the 3 cross sections just some hours (<4 h) apart, you get a snapshot of an existing dust plume whose maximum is currently over the central Atlantic. Lower values of the backscatter coefficient above the Sahara and the Caribbean (cross section 1 and 3) can not be directly linked to emission and deposition (named by you "emission phase" and "deposition phase"). Usually, there are several days between emission and deposition and not just some hours. So, there is no benefit in reporting the backscatter values for the 3 cross sections. I would consider removing these values from the abstract and the conclusion.

   Your next Section 4.3 is better suited to follow the dust from emission to deposition.

AR: Thanks for the suggestion. Yes, we agree with you. We think **the dust layers captured by Aeolus and CALIPSO during several hours on 19 June 2020 (cross-section 1, 2 and 3 in Fig 6 (b), (c)) are relatively static compared with the whole dust plume transport process. Namely, we took a snapshot of the dust plumes on this day.** As discussed in Section 4.1 and Section 4.3 of the manuscript, the emission-transport-deposition process of the dust plume needs almost two weeks, not just some hours.

   Sorry for the misleading. **"During emission phase", "during development phase" and "during deposition phase" in the manuscript have been modified as "over the emission region (Western Sahara)", "over the transport region (Middle Atlantic)" and "over the deposition region (Western Atlantic)"** in the description of the dust

advection values on 19 June 2020. Besides, **we rewrote the part of Section 4.2, which** **is renamed as "Observation snapshot of the dust plume and dust advection** **calculation on 19 June 2020", to illustrate the overall geographical distribution of** **dust layers as a snapshot on the morning of this day**. And **the mean backscatter** **values of the 3 cross sections have been removed from the abstract and the** **conclusion**. The revised descriptions of this portion are shown as below:

[revised manuscript text omitted]

2. The calculation of the mean mass concentration is not well defined. How do you define your dust layer? Or do you take an average over the whole cross section? You mention some upper and lower threshold values for the mass concentration based on previous observations. However, if you observe such an intense dust event ("Godzilla"), the mass concentration may exceed the upper threshold. To calculate a mean mass concentration, you should define your dust layer, probably with a

lower backscatter or extinction coefficient threshold and then take the average over the entire dust layer.

AR: **Actually, the dust aerosol was identified and verified by two steps.** Firstly, before the particle mass concentration estimation, **the Vertical Feature Mask (VFM) product from CALIOP was used to identify the dust aerosol.** Only the data bins (from the common data pixel grid of Aeolus and CALIPSO) identified as "dust" are applied in the estimation of the dust mass concentration. Secondly, the **relative humidity data provided by ECMWF is used to filtrate dust aerosol which has absorbed moisture.** When the relative humidity is larger than 90%, the dust aerosol will be influenced by the hygroscopicity effect and its properties could change. Then the mass concentration calculation method does not make sense any more. After two steps of dust aerosol identification and verification, the "real" dust aerosol was selected and its optical properties (backscatter coefficients at 532 nm and 1064 nm, extinction coefficients at 355 nm, 532 nm and 1064 nm) are used in the estimation of the dust mass concentration. **In conclusion, we did not take an average over the whole cross section, but the filtered cross section instead.**

Thanks for the suggestion. **We recalculated the mean mass concentration of each dust layer with a lower mass concentration threshold and without an upper mass concentration threshold.**
It is positive, that you compare two different methods. In order to judge the differences, you should add uncertainties to both derived mean mass concentrations (Table 1+2).
AR: Thanks for the suggestion. We have added the uncertainties to both mean mass concentrations in the Table 1 and Table 2 of the revised manuscript, which are shown as below:

**"Table 1. Mean dust mass concentration of each cross-sections on 19 June 2020 calculated by two methods**

| Cross-section | 1 | 2 | 3 |
| --- | --- | --- | --- |

| | | | |
|---|---|---|---|
| Mean mass concentration, $\mathrm{mg \cdot m^{-3}}$ (the retrieval method) | $0.28 \pm 0.23$ | $0.26 \pm 0.24$ | $0.22 \pm 0.19$ |
| Mean mass concentration, $\mathrm{mg \cdot m^{-3}}$ (the factor method) | $0.37 \pm 0.24$ | $0.40 \pm 0.25$ | $0.39 \pm 0.27$ |

**Table 2. Mean dust mass concentration of each cross-sections at different times during the dust transport calculated by two methods**

| Date | 15 June | 16 June | 19 June | 24 June | 27 June |
|---|---|---|---|---|---|
| Mean mass concentration, $\mathrm{mg \cdot m^{-3}}$ (retrieval method) | $0.30 \pm 0.23$ | $0.27 \pm 0.24$ | $0.26 \pm 0.24$ | $0.27 \pm 0.24$ | $0.22 \pm 0.19$ |
| Mean mass concentration, $\mathrm{mg \cdot m^{-3}}$ (factor method) | $0.26 \pm 0.17$ | $0.39 \pm 0.24$ | $0.40 \pm 0.25$ | $0.42 \pm 0.21$ | $0.34 \pm 0.20$ |

"

For the factor method, do you use the extinction coefficient provided by CALIPSO or the extinction coefficient calculated with the adapted lidar ratio (58 sr)? The later would be preferable to be consistent with your advection calculation procedure.

AR: Thanks. We used the extinction coefficient calculated with the adapted lidar ratio (58 sr) in the factor method.

3. In Section 4.3, you should make sure that the same dust was observed in all the cross sections. The description stays a bit vague. A so-called Lagrangian case study was presented in Weinzierl et al., BAMS 2017, there an aircraft observed the same dust sample at the coast of Africa and some days later over the Caribbean. You have

all the trajectory calculations ready, just use them in a more quantitative way to show that you track the same dust event. For example, you could add dots to the trajectories marking intervals of 24 h in Fig. 9. The dots alone won't be sufficient.

AR: Thanks for the suggestion. Please be aware that, since we omitted the Figure 7 and 10 in the revised manuscript, the original Figure 9 becomes Figure 8. We modified Fig. 8(a) and (d) in the revised manuscript. **The square symbols on the HYSPLIT trajectory lines have been added in the revised Fig. 8(a) and (d) to indicate the trajectories' locations of 15 June, 16 June, 19 June, 24 June and 27 June, which are matched with the 5 cross-sections by ALADIN and CALIOP.**

[revised manuscript text omitted]

4. The CALIPSO examples introduced in Fig. 4 and 5 are later on not used anymore. It would be better to show in Fig. 4 some dates used in Section 4.3. In Fig. 5 you should definitely show the case of 19 June 2020 because it is later on used in the case study of Section 4.2.

AR: Thanks for the suggestions. We have replaced the VFM data and the corresponding CALIOP scanning tracks on 18 June 2020 and 23 June 2020 with those on 16 June 2020 and 27 June 2020 in the revised Fig. 4. The profiles and the trajectories in the revised Fig. 5 have been replaced as the case of 19 June 2020. The revised Fig. 4 and Fig. 5 are shown as below:

"

[Figure]

**Figure 4. Vertical feature mask from CALIPSO L2 product (a) on 16 June 2020 over the west coast of Africa and the eastern Atlantic and (b) on 27 June 2020 over the western Atlantic (around the east coast of America). (c) and (d) show the corresponding CALIOP scanning tracks of (a) and (b) respectively, the arrows in which indicate the motion direction of CALIPSO (https://www-calipso.larc.nasa.gov/products/lidar/browse_images/production/, last access: 24 March 2022)."**

"

[Figure]

**Figure 5. (a)(c)(e) CALIPSO total backscatter coefficient profiles and particle depolarization ratio profiles capturing dust layers at around 0400UTC 19 June 2020. (b)(d)(f) HYSPLIT backward trajectories and forward trajectories at different sites of corresponding CALIPSO profiles and different heights on 0400UTC 19 June 2020. The backward and forward trajectories' durations are 120 hours and 192 hours respectively (https://www.ready.noaa.gov/hypub-bin/trajtype.pl?runtype=archive, last access: 23 March 2022)."**

5.  It is a great step forward to use the lidar ratios for (Western) Saharan dust instead of global averages. The lidar ratio of 60 sr at 1064 nm seems a good estimate as recently confirmed by Haarig et al., ACP 2022 (57 – 69 sr). Although, a higher ratio of LR1064/LR532 was reported. Nevertheless, the values used seem to be reasonable.

AR: Thanks.

6. Aeolus aerosol products are usually reported on a very coarse horizontal resolution. How do you make sure that your profiles are not influenced by clouds? You are talking about the cloud screening in the case of CALIPSO, but not for Aeolus. Please add some comments on the cloud and aerosol separation in the case of Aeolus.

AR: Firstly, we **set strict match criterions of the ALADIN and the CALIOP scanning tracks**: (1) The distances between two satellites scanning tracks are less than 200 km; (2) The tracks of Aeolus are downwind of the tracks of CALIPSO. Secondly, **we utilized wind field data and relative humidity data from ECMWF as auxiliary data to illustrate the homogeneity between the matched two spaceborne lidars' scanning tracks**. Because of the relatively short distances and the stable wind fields (both of the standard deviation percentages of wind speed and direction between the tracks along each latitude line are less than 10%) between the matched tracks, it is considered that **the atmospheric conditions and the aerosol types are approximately the same on both two spaceborne lidars' scanning tracks**. Therefore, in the common data pixel grid of the Aeolus data and the CALIPSO data, **the cloud screening and the dust aerosol selection of CALIPSO are also approximatively suitable to the Aeolus data**. Besides, if relative humidity is larger than 94%, then the probability that cloud presents is quite high (Flament et al., 2021). **Before the estimation of dust mass concentration, the Aeolus data is filtered when the relative humidity is larger than 90%**, which can also support to screen possible cloud conditions in the case of Aeolus data.

*Reference: Flamant, P., Dabas, A., Martinet, P., Lever, V., Flament, T., Trapon, D., Olivier, M., Cuesta, J., and Huber, D.: Aeolus L2A Algorithm Theoretical Baseline Document, Particle optical properties product, version 5.7, available at: https://earth.esa.int/eogateway/catalog/aeolus-l2a-aerosol-cloud-optical-product (last access: 15 March 2022), 2021*

7. Please add uncertainties to all your calculated values, especially to the mean dust advection values. Otherwise, you can't draw conclusions on changing values.

AR: The uncertainties of each cross-sections' mean mass concentration are added, and presented in Table 1 and 2 in the revised manuscript:

"**Table 1. Mean dust mass concentration of each cross-sections on 19 June 2020 calculated by two methods**

| Cross-section | 1 | 2 | 3 |
|---|---|---|---|
| Mean mass concentration, $mg \cdot m^{-3}$ (the retrieval method) | $0.28 \pm 0.23$ | $0.26 \pm 0.24$ | $0.22 \pm 0.19$ |
| Mean mass concentration, $mg \cdot m^{-3}$ (the factor method) | $0.37 \pm 0.24$ | $0.40 \pm 0.25$ | $0.39 \pm 0.27$ |

**Table 2. Mean dust mass concentration of each cross-sections at different times during the dust transport calculated by two methods**

| Date | 15 June | 16 June | 19 June | 24 June | 27 June |
|---|---|---|---|---|---|
| Mean mass concentration, $mg \cdot m^{-3}$ (retrieval method) | $0.30 \pm 0.23$ | $0.27 \pm 0.24$ | $0.26 \pm 0.24$ | $0.27 \pm 0.24$ | $0.22 \pm 0.19$ |
| Mean mass concentration, $mg \cdot m^{-3}$ (factor method) | $0.26 \pm 0.17$ | $0.39 \pm 0.24$ | $0.40 \pm 0.25$ | $0.42 \pm 0.21$ | $0.34 \pm 0.20$ |

"

The uncertainties of the mean dust advection values are also added in the revised manuscript, which are shown as below:

"On 19 June, the mean dust advection values are about $1.91 \pm 1.21 \, mg \cdot m^{-2} \cdot s^{-1}$ over the emission region, $1.38 \pm 1.28 \, mg \cdot m^{-2} \cdot s^{-1}$ over the transport region and $0.75 \pm 0.68 \, mg \cdot m^{-2} \cdot s^{-1}$ over the deposition region. In the whole life-time of the dust event,

the mean dust advection values are about $1.51\pm1.03\ \mathrm{mg\cdot m^{-2}\cdot s^{-1}}$ on 15 June 2020, $2.19\pm1.72\ \mathrm{mg\cdot m^{-2}\cdot s^{-1}}$ on 16 June 2020, $1.38\pm1.28\ \mathrm{mg\cdot m^{-2}\cdot s^{-1}}$ on 19 June 2020, $1.60\pm1.08\ \mathrm{mg\cdot m^{-2}\cdot s^{-1}}$ on 24 June 2020 and $1.03\pm0.60\ \mathrm{mg\cdot m^{-2}\cdot s^{-1}}$ on 27 June 2020." (from Section 5 of the revised manuscript)

Minor comments

8. Text insides some figures (especially Fig. 3 + 4) is quite small and hard to read.

AR: Thanks. Figure 3 and 4 have been modified in the revised manuscript as below:

[Figure]

"

**Figure 3. The Dust Score Index provided by AIRS/Aqua at different stages, including emission, transportation, dispersion and deposition (https://airs.jpl.nasa.gov/map/, last access: 10 January 2022).**

[Figure]

**Figure 4. Vertical feature mask from CALIPSO L2 product (a) on 16 June 2020 over the west coast of Africa and the eastern Atlantic and (b) on 27 June 2020 over the western Atlantic (around the east coast of America). (c) and (d) show the corresponding CALIOP scanning tracks of (a) and (b) respectively, the arrows in which indicate the motion direction of CALIPSO (https://www-calipso.larc.nasa.gov/products/lidar/browse_images/production/, last access: 24 March 2022)."**

9. Figure 7 and 10 are quite complex and hard to follow. The text is understandable even without these figures. In case of the wind speed and direction, you have the nice Fig. 12, and the other information from Fig. 7 and 10 are not necessary to understand the paper. I would consider removing these figures to make the paper easier to read.

AR: Thanks for your comments. We removed the original Fig. 7 and Fig. 10 in the revised manuscript. The corresponding explanations regarding the smooth distribution of wind fields and RH are also stated in the revised manuscript. Hence, please be aware that the figure numbers are changed accordingly.

10. L55: A reference about SHADOW is missing. What about Veselovskii et al., ACP 2016?

AR: Sorry for the careless. The reference Veselovskii et al. (2016) has been added in the revised manuscript:

Veselovskii, I., Goloub, P., Podvin, T., Bovchaliuk, V., Derimian, Y., Augustin, P., Fourmentin, M., Tanre, D., Korenskiy, M., Whiteman, D. N., Diallo, A., Ndiaye, T.,

Kolgotin, A., and Dubovik, O.: Retrieval of optical and physical properties of African dust from multiwavelength Raman lidar measurements during the SHADOW campaign in Senegal, Atmos. Chem. Phys., 16, 7013–7028, https://doi.org/10.5194/acp-16-7013-2016, 2016.

11. The technical details about Aeolus could be moved from the introduction to Section 2.1. Just keep the most important facts about Aeolus as you have done it for CALIPSO.

AR: Thanks for the suggestion. We have moved the technical details about Aeolus from introduction to Section 2.1.

The revised description about Aeolus in the introduction are shown as below:

"…Thanks to the efforts of the European Space Agency (ESA), a first ever spaceborne direct detection wind lidar, Aeolus, which is capable of providing vertical wind fields globally with high temporal and spatial resolution has been developed under the framework of the Atmospheric Dynamics Mission (ADM) (Stoffelen et al., 2005; ESA, 1999; Reitebuch et al., 2012; Kanitz et al., 2019). The Atmospheric Laser Doppler Instrument (ALADIN) is a direct detection high spectral resolution wind lidar carried by Aeolus and provides the vertical profiles of the Horizontal-Line-of-Sight (HLOS) wind speeds. Further, the wind vector data assimilated with the HLOS wind speed data and the particle optical property data (e.g., extinction coefficient, backscatter coefficient) at 355 nm are also provided in the products of Aeolus."

The technical details about Aeolus in the revised Section 2.1 of the manuscript are shown as below:

"**2.1 ALADIN/Aeolus**

On 22 August 2018, Aeolus was successfully launched into its sun-synchronous orbit at a height of 320 km (Witschas et al., 2020; Lux et al., 2020). A quasi-global coverage is achieved daily (~ 15 orbits per day) and the orbit repeat cycle is 7 days (111 orbits). The orbit is sun-synchronous with a local equatorial crossing-time of ~ 6 am/pm. ALADIN, which is the unique payload of Aeolus, is a direct detection high spectral resolution wind lidar. It is a pulsed ultraviolet lidar working at the wavelength of 354.8 nm with a laser pulse energy around 65 mJ and with a repetition of 50.5 Hz. As the receiver, a 1.5 m diameter telescope collects the backscattered light. In order to retrieve the LOS wind speeds, the Doppler shifts of light caused by the motion of molecules and aerosol particles need to be identified. Aiming at this, a Fizeau interferometer is applied in the Mie channel to extract the frequency shift of the narrow-band particulate return signal by means of the fringe imaging technique (Mckay, 2002). In the Rayleigh channel, two coupled Fabry-Perot interferometers are used to analyze the frequency shift of the broad-band molecular return signal by the double edge technique (Chanin et al., 1989; Flesia and Korb, 1999). …"

12. 4 VFM – please write vertical feature mask

AR: Thanks, revised.

13. 4 "west coast of Africa"

AR: Thanks, revised.

14. 5 The term "source" might be misleading, because you show a "position" along the CALIPSO track and the corresponding profiles at this position. And then you use this position as source for your trajectories. Reading "source" reminded me on dust sources.

AR: Sorry for the misleading and thanks for the suggestion. We replaced "source" with "position" in Fig. 5 and the relevant description in Section 4.1 of the revised manuscript.

15. 6a – it is not a "vertical" view and HYSPLIT trajectories are not shown.

AR: Thanks. The caption of Fig. 6 (a) has been revised as "(a) Aeolus and CALIPSO scanning tracks".

16. L286 Explain u and v component of wind vector to readers not familiar with these conventions.

AR: Thanks. The sentence has been revised as "The zonal wind velocity (u component of the wind vector, from west point to east), meridional wind velocity (v component of the wind vector, from south point to north) and supplementary geophysical parameters are contained in L2C data product." in the revised manuscript.

17. L310 "Godzilla" – a nice piece of information which could already be placed in the introduction.

AR: Thanks. The relevant information about "Godzilla" has been added in the front of the last paragraph of Section 1 in the revised manuscript, which is also shown as below:

   "A long-term, large-scale Sahara dust transport event which occurred between 14 June and 27 June 2020 is captured, tracked and analyzed. Because of this record-breaking trans-Atlantic African dust plume, the magnitude and duration of spaceborne-sensors retrieved aerosol optical depth over the tropical North Atlantic Ocean were the greatest ever observed during summer over the past 18 years (Pu and Jin, 2021). This dust plume caused a historic, massive African dust intrusion into the Caribbean Basin and southern US, which is nicknamed the "Godzilla" dust plume (Yu et al., 2021)."

AR: In Fig. 7 (corresponding to the Figure 8 in the original manuscript), the color plots are shown exactly on the middle of the CALIPSO tracks and the Aeolus tracks, to represent the dust advection over the region between two satellites' tracks. The modified Fig. 7 are shown as below:

[Figure]

"

Figure 7. The dust advection calculated with data from ALADIN, CALIOP and ECMWF (a) the dust advection values at different cross-sections of dust plumes and (b) the dust advection directions at different cross-sections of dust plumes on 19 June 2020."

19. L341 "dust mass" – you're not showing the dust mass, but "enhanced backscatter and extinction values indicating the presence of dust"

AR: Thanks. This sentence has been revised as "From these figures, we can find that at different cross-sections of Aeolus and CALIPSO, the dust transport modelled with HYSPLIT match well with the enhanced backscatter and extinction coefficient values indicating the presence of dust." in the revised manuscript.

20. 9d It is almost impossible to capture the latitudinal component in the plot – I would consider to show it on altitude – longitude plane (this is the interesting information!) and indicate the different positions in latitude by different lines, e.g., position A in dashed lines, position B in dotted lines, …

AR: Thanks. We modified Fig. 8(d) (corresponding to Figure 9 of the original manuscript) according to your suggestions. The modified Fig. 8(d) are shown as below:

[Figure]

We show it on altitude-longitude panel. The different colors of the lines indicate the HYSPLIT trajectories modelled from different positions (the red lines are from position A, the magenta lines are from position B, the orange lines are from position C). the different styles of the lines indicate the HYSPLIT trajectories modelled from different altitude (the solid lines are from 3 km, the dot lines are from 4 km, the dot dash lines are from 5 km). The squares on the HYSPLIT trajectory lines have been added to indicate the trajectories position of 15 June, 16 June, 19 June, 24 June and 27 June, which are matched with the 5 cross-sections by ALADIN and CALIOP.

21. The "Saharan dust westward transport tunnel" (L.383) is somehow linked to the "Saharan Air Layer".

AR: Thanks for the reminder. We revised the conclusion of the wind field cross-sections of this dust transport event observed by Aeolus as "Therefore, it can be considered that Aeolus provided the observations of the dynamics of this dust transport event in the Saharan air Layer (SAL), which is a hot, dry, elevated layer originating from the Sahara Desert and covering large parts of the tropical Atlantic (Carlson and Prospero, 1972; Prospero and Carlson, 1972)." And the references are added as:

Carlson, T. N., and Prospero, J. M.: The Large-Scale Movement of Saharan Air Outbreaks over the Northern Equatorial Atlantic, Journal of Applied Meteorology and Climatology, 11(2), 283-297. https://doi.org/10.1175/1520-0450(1972)011<0283:TLSMOS>2.0.CO;2, 1972.

Prospero, J. M., and Carlson, T. N.: Vertical and areal distribution of Saharan dust over western equatorial north Atlantic Ocean, J. Geophys. Res., 77, 5255–5265, doi:10.1029/JC077i027p05255, 1972.

---

## Author Comment (AC2)

This paper describes the combination of satellite borne lidar data from two instruments in combination with model wind field data and back and forward trajectory analyses to investigate the advection of a major dust storm across the Atlantic Ocean. The paper focus on a case study of a major dust storm to assess how the combined CALIPSO and Aeolus satellite products can be combined with ECMWF driven trajectories to describe dust transport and loss. The paper was previously submitted to ACPD and this version has been considerably improved.

I do, however, have a major reservation about section 4.2 and the accompanying statements in the abstract and summary sections. Section 4.2 presents lidar curtains at three locations across the sub-tropical north Atlantic on a day in the middle of the dust storm. The three curtains are close to the source region, over the mid-Atlantic and towards the west, in the far-field of the plume. However, the satellite overpasses presented are taken only 3 hours apart. The advection times between the most easterly lidar curtain and the most westerly are of the order of a week or more. The data in section 4.2 show the overall geographical distribution of dust across the Atlantic as a snapshot on the morning of 19/6/2020. What they do not do is say anything at all about the dynamics of the dust plume as it advects across the Atlantic region. The source region may have changed or emissions of dust varied and the transport pathways may be affected by changing atmospheric conditions over the course of the event. However, section 4.2 assumes the dust plume is time invariant and describes the scene as representing different ages of the plume. This is misleading and in any case is described much better in section 4.3. Either section 4.2 should be rewritten to illustrate geographical variability at a single point in time or removed. Furthermore, the way the results from this section are presented in the abstract and summary should be reframed or removed as they are written as though the data were taken in a pseudo lagrangian way and they were not.

AR: Thanks for the suggestion. Actually, we also think **the dust layers captured by Aeolus and CALIPSO during several hours on 19 June 2020 (cross-section 1, 2**

**and 3 in Fig 6 (b), (c)) are relatively static compared with the whole dust plume transport process.** Sorry for the misleading. According to your suggestion, **we rewrote the part of Section 4.2 and reframed the relevant conclusion in the abstract and the summary. Section 4.2 of the revised manuscript has been renamed as "Observation snapshot of the dust plume and dust advection calculation on 19 June 2020".** The description was reframed to **illustrate the overall geographical distribution of dust layers as a snapshot on the morning of this day**. The revised part of Section 4.2 is shown as below:

"**4.2 Observation snapshot of the dust plume and dust advection calculation on 19 June 2020**

In this section, the dust event observation snapshot captured by ALADIN and CALIOP on 19 June 2020 is introduced in detail. The quasi-synchronized observations from ALADIN and CALIOP on 19 June 2020 are presented in Fig. 6, where the purple lines indicate the scanning tracks of ALADIN and the green lines indicate the scanning tracks of CALIOP. It is found that the overpasses of each satellite are only around 3 hours apart. Hence, we captured the dust layers on the morning of 19 June 2020 quasi-simultaneously over the Western Sahara, the Middle Atlantic and the Western Atlantic, i.e., took a snapshot of the dust plumes. From the profiling of dust optical properties, discriminated by the CALIOP measurements, the dust geographical distribution over Atlantic Ocean on this day could be determined. The extinction coefficients and backscatter coefficients at the wavelengths of 355 nm, 532 nm and 1064 nm within the dust mass are also determined. From the profiling, it was found that the mean backscatter coefficients at 532 nm were about $3.88 \times 10^{-6} \pm 2.59 \times 10^{-6} \text{ m}^{-1}\text{sr}^{-1}$ in "cross-section 1", $7.09 \times 10^{-6} \pm 3.34 \times 10^{-6} \text{ m}^{-1}\text{sr}^{-1}$ in "cross-section 2" and $7.76 \times 10^{-6} \pm 3.74 \times 10^{-6} \text{ m}^{-1}\text{sr}^{-1}$ in "cross-section 3". On 19 June 2020, the dust layers existed over the Western Sahara, the Middle Atlantic and the Western Atlantic quasi-simultaneously, which indicates that the dust plume area over the Atlantic on the morning of this day is quite enormous and this dust transport event is massive and

extensive.

……

In Fig. 7, the dust advection at different heights of the three cross-sections are presented. From the profiling, the mean dust advection value is about $1.91 \pm 1.21 \, \mathrm{mg \cdot m^{-2} \cdot s^{-1}}$ in "cross-section 1" (over the emission region), $1.38 \pm 1.28 \, \mathrm{mg \cdot m^{-2} \cdot s^{-1}}$ in "cross-section 2" (over the transport region) and $0.75 \pm 0.68 \, \mathrm{mg \cdot m^{-2} \cdot s^{-1}}$ in "cross-section 3" (over the deposition region), respectively. In conclusion, on 19 June 2020, the dust layers over the Western Sahara, the Middle Atlantic and the Western Atlantic are observed by ALADIN and CALIOP nearly in the meanwhile. And the dust advections of the three cross-sections indicate the quasi-simultaneous transport of the dust plumes over the emission region, the transport region and the deposition region on the same day."

The revised parts of the abstract and the summary are shown as below:

"…From the measurement results on 19 June 2020, the dust plumes are captured quasi-simultaneously over the emission region (Western Sahara), the transport region (Middle Atlantic) and the deposition region (Western Atlantic) individually, which indicates that the dust plume area over the Atlantic on the morning of this day is quite enormous and this dust transport event is massive and extensive. The quasi-synchronization observation results of 15, 16, 19, 24 and 27 June by ALADIN and CALIOP during the entire transport process show good agreement with the "Dust Score Index" data and the HYSPLIT trajectories, which indicates that the transport process of the same dust event is tracked by ALADIN and CALIOP, verifies that the dust transport spent around 2 weeks from the emission to the deposition and achieved the respective observations of this dust event's emission phase, development phase, transport phase, descent phase and deposition phase. Finally, the advection value for different dust parts and heights on 19 June and on the entire transport routine during transportation are

computed. On 19 June, the mean dust advection values are about $1.91 \pm 1.21 \, \text{mg} \cdot \text{m}^{-2} \cdot \text{s}^{-1}$ over the emission region, $1.38 \pm 1.28 \, \text{mg} \cdot \text{m}^{-2} \cdot \text{s}^{-1}$ over the transport region and $0.75 \pm 0.68 \, \text{mg} \cdot \text{m}^{-2} \cdot \text{s}^{-1}$ over the deposition region." (from the abstract)

"…From the measurement results on 19 June 2020, the dust plumes are captured quasi-simultaneously over the emission region (Western Sahara), the transport region (Middle Atlantic) and the deposition region (Western Atlantic) individually, which indicates that the dust plume area over the Atlantic on the morning of this day is quite enormous and this dust transport event is massive and extensive. The quasi-synchronization observation results of 15, 16, 19, 24 and 27 June by ALADIN and CALIOP during the entire transport process show good agreement with the "Dust Score Index" data and the HYSPLIT trajectories, which indicates that the transport process of the same dust event is tracked by ALADIN and CALIOP, verifies that the dust transport spent around 2 weeks from the emission to the deposition and achieved the respective observations of this dust event's emission phase, development phase, transport phase, descent phase and deposition phase.

Finally, the advection at different dust parts and heights on 19 June and on the entire transport routine during transportation are computed, respectively. On 19 June, the mean dust advection values are about $1.91 \pm 1.21 \, \text{mg} \cdot \text{m}^{-2} \cdot \text{s}^{-1}$ over the emission region, $1.38 \pm 1.28 \, \text{mg} \cdot \text{m}^{-2} \cdot \text{s}^{-1}$ over the transport region and $0.75 \pm 0.68 \, \text{mg} \cdot \text{m}^{-2} \cdot \text{s}^{-1}$ over the deposition region, from which we can infer the quasi-simultaneous transport of the dust plumes over the emission region, the transport region and the deposition region on this day…" (from the summary)

Specific recommendations

Lines 62-63: "Additionally, the CALIOP product Vertical Feature Mask product (VFM)" better to write

"Additionally, the CALIOP Vertical Feature Mask product (VFM)"

AR: Thanks, it is revised.

Line 74 "(e)motion"

AR: Thanks, it is revised.

Line 170-174 "Based on the dataset consists of the backscatter coefficients and extinction coefficients at the wavelengths of 1064 nm and 532 nm from CALIOP and the extinction coefficients at the wavelength of 355 nm from ALADIN, the aerosol volume concentration distribution can be calculated based on the regularization method which was performed by generalized cross-validation (GCV) from Müller et al. (1999)." A confusing sentence that needs to be rewritten

AR: This sentence has been rewritten as "Based on the dataset consisting of the backscatter coefficients and extinction coefficients at the wavelengths of 1064 nm and 532 nm from CALIOP and the extinction coefficients at the wavelength of 355 nm from ALADIN, the aerosol volume concentration distribution can be estimated based on the regularization method which was performed by generalized cross-validation (GCV) from Müller et al. (1999)."

lines 240-241: Figure 4a shows the majority of the dust has been lifted to a maximum of around 7km or less south of 20N on 18/6/2020, there is only a small proportion of the dust at the far north end of the overpass that has a maximum close to 10 km.   This probably needs rephrasing.

AR: Thanks for the suggestion. We updated Fig. 4 with the VFM products on 16 June 2020 and 27 June 2020, to make them matched with part of the satellite cross-sections presented in Section 4.3. The modified Fig. 4 and the relevant description are shown as below:

"Figure 4 presents the vertical distribution of the dust plume during the development phase (16 June 2020) over the eastern Atlantic and during the deposition phase (27 June 2020) over the western Atlantic. From Fig. 4 (a), it can be seen that the dust plume has been lifted up to around 7 km. Figure 4 (b) presents the descending dust plume, the bottom of which may mix with marine aerosol and become dusty marine aerosol. Therefore, the VFM data of CALIPSO captures the dust plume vertically over the eastern and the western Atlantic and verifies the dust transportation process.

[Figure]

**Figure 4. Vertical feature mask from CALIPSO L2 product (a) on 16 June 2020 over the west coast of Africa and the eastern Atlantic and (b) on 27 June 2020 over the western Atlantic (around the east coast of America). (c) and (d) show the corresponding CALIOP scanning tracks of (a) and (b) respectively, the arrows in which indicate the motion direction of CALIPSO (https://www-calipso.larc.nasa.gov/products/lidar/browse_images/production/, last access: 24 March 2022)."**

Lines 276-282: The narrative in the section assumes a pseudo-langragian language but the lidar passes are on the same day so these are different slices of a dust event that has lasted several days (fig 3) and has a transit time of multiple days between the overpasses shown in fig 5. The wording here needs to better reflect that these are cross sections at different geophysical locations in the plume and do not directly represent plume evolution. This discussion is extended to report values of backscatter and advection for different phases of the dust plume. However, these don't reflect actual advection of the same air. The underlying assumption is the dust plume does not change with time. Clearly, this is not the case, so the determinations from the 3 different overpasses cant really be compared in the way that is done in the analysis in 4.2. At best this gives a snapshot of the plume at a single point in time across much of the Atlantic. This section needs to be rewritten in my view to make this clear and to convey why this is appropriate, otherwise it is best removed. This same approach is also followed up in the summary (402-406). The analysis is not pseudo-lagrangian and should not be inferred as such, the different phases of the storm were emitted many days apart and may have had very different conditions at source and during advection. This needs to be made explicit. The abstract also has the same errors between lines 22-25. This needs to be removed or corrected.

AR: Thanks for the suggestion. **We reframed and rewrote part of Section 4.2 and the relevant conclusion in the abstract and the summary to illustrate the overall geographical distribution of dust layers as a snapshot on the morning of this day.**

The revised part of Section 4.2 is shown as below:

[revised manuscript text omitted]

Line 293: "to calculate(d)"

AR: Thanks, it is revised.

Line 384: Affected not effected

AR: Thanks, it is revised.